# Targeted delivery of Fc-fused PD-L1 for effective management of acute and chronic colitis

Xudong Tang [1,4], Yangyang Shang [1,2,4], Hong Yang[1,4], Yalan Song[1], Shan Li[1], Yusi Qin[1], Jingyi Song[1], Kang Chen[1], Yang Liu[3], Dinglin Zhang [2] ✉ & Lei Chen [1] ✉

The PD-1/PD-L1 pathway in mucosal immunity is currently actively explored and considered as a target for inflammatory bowel disease (IBD) treatment. However, systemic PD-L1 administration may cause unpredictable adverse effects due to immunosuppression. Here we show that reactive oxygen species (ROS)-responsive nanoparticles enhance the efficacy and safety of PD-L1 in a mouse colitis model. The nanoparticles control the accumulation and release of PD-L1 fused to Fc (PD-L1-Fc) at inflammatory sites in the colon. The nanotherapeutics shows superiority in alleviating inflammatory symptoms over systemic PD-L1-Fc administration and mitigates the adverse effects of PD-L1-Fc administration. The nanoparticles-formulated PD-L1-Fc affects production of proinflammatory and anti-inflammatory cytokines, attenuates the infiltration of macrophages, neutrophils, and dendritic cells, increases the frequencies of Treg, Th1 and Tfh cells, reshapes the gut microbiota composition; and increases short-chain fatty acid production. In summary, PD-L1-Fc-decorated nanoparticles may provide an effective and safe strategy for the targeted treatment of IBD.

The incidence of inflammatory bowel disease (IBD) in Asia has increased in recent years, and patients' quality of life is seriously affected. The pathogenesis of IBD is considered to be related to heredity, environmental factors, immunity and the intestinal flora[1]. Among these factors, disorders related to autoimmunity in the intestines have an extremely important role in the pathogenesis of IBD[2–4]. The clinical symptoms of IBD patients are mainly caused by intestinal inflammation. Conventional anti-inflammatory drugs have been used for four decades, but the therapeutic effect is not always satisfactory. Therefore, an increasing number of studies have focused on biotherapies for IBD, which may alleviate intestinal inflammation by inhibiting different inflammatory factors. Although the effectiveness of biotherapy has been confirmed since its clinical application, the associated adverse reactions and safety issues have also attracted attention[5–7].

The programmed cell death protein-1 (PD-1)/programmed cell death-ligand (PD-L1) pathway plays an important role in maintaining peripheral immune tolerance and ameliorating autoimmune diseases by preventing excessive inflammatory reactions and corresponding immune damage[8–11]. PD-L1 expressed in the colon suppresses the proliferation of activated CD4[+] and CD8[+] effector T cells and plays a prominent role in mucosal tolerance to commensal bacteria and dietary antigens[12]. PD-1 inhibitors (e.g., nivolumab, pembrolizumab and avelumab) and PD-L1 inhibitors (e.g., atezolizumab) have been reported to induce immune-related colitis[13,14]. The expression of PD-L1 in the intestinal mucosa of patients with IBD or immune-associated

[1]Institute of Gastroenterology of PLA, Southwest Hospital, Army Medical University (Third Military Medical University), Chongqing 400038, China. [2]Department of Chemistry, College of Basic Medicine, Army Medical University (Third Military Medical University), Chongqing 400038, China. [3]Department of Laboratory Animal Science, College of Basic Medicine, Army Medical University (Third Military Medical University), Chongqing 400038, China. [4]These authors contributed equally: Xudong Tang, Yangyang Shang, Hong Yang. ✉e-mail: zh18108@tmmu.edu.cn; chenlei_1977603@126.com

colitis is higher than that in normal tissues. Blocking the PD-1/PD-L1 pathway can result in different degrees of inflammatory lesions in the colon, whereas PD-L1 administration can relieve symptoms of enteritis in dextran sulfate sodium (DSS)-induced colitis mice[15–19]. Thus, PD-L1 administration may be an optional approach for the treatment of IBD. However, PD-L1 has a strong immunosuppressive effect, and systemic administration of PD-L1 is associated with increased risks of infection and carcinogenesis. Therefore, accurate delivery of PD-L1 to the inflamed intestinal mucosa needs to be further investigated.

Previously, we synthesized 4-phenylboronic acid pinacol ester-conjugated cyclodextrin biomaterials (Oxi-αCD), which are reactive oxygen species (ROS)-responsive nanomaterials with favorable biocompatibility and biosecurity that can serve as drug carriers for the targeted treatment of melanoma and rheumatoid arthritis[20,21]. Given that ROS overproduction is closely related to the pathogenesis of IBD[22], the ROS-responsive nanomaterials may also be useful for drug delivery at inflamed sites of intestine. We hypothesized that the ROS-responsive nanoparticles would arrive to the inflamed sites of colon through the vascular system and, upon subsequent hydrolyzation by high concentrations of ROS, leading to the local release of their cargo with immune modulatory functions.

Here we show the ROS-responsive nanoparticles carrying the Fc-fused PD-L1 for the targeted treatment of colitis (named PD-L1-Fc/Oxi-αCD nanoparticles) can effectively deliver PD-L1-Fc to the inflamed colon and obviously alleviate the symptoms of acute and chronic colitis by regulating the immune cells/cytokines associated with inflammation, as well as the microbial community and SCFA metabolism in the colon. The good therapeutic effect and favorable biosafety of the PD-L1-Fc/Oxi-αCD nanoparticles in DSS-induced mouse models establish confidence for the further development of targeted therapy for colitis. Therefore, the PD-L1-Fc/Oxi-αCD nanoparticles have the potential to be an alternative therapeutic approach for IBD treatment.

## Results

### Construction and characterization of the PD-L1-Fc-loaded nanoparticles

The HPAP-conjugated α-CD (Oxi-αCD) material used to fabricate ROS-responsive nanoparticles was synthesized and verified by proton nuclear magnetic resonance ($^1$H-NMR) in our previous studies[21,23]. Oxi-αCD nanoparticles were fabricated using a nanoprecipitation/self-assembly method (defined as Blank/Oxi-αCD nanoparticles). The periphery of the Blank/Oxi-αCD nanoparticles is coated with lecithin and DSPE-PEG$_{2000}$-NHS, which helps to stabilize the nanoparticles and prolong their circulation time in the blood. In addition, DSPE-PEG$_{2000}$-NHS can form amide bonds with the PD-L1-Fc protein to fabricate PD-L1-Fc-decorated nanoparticles (defined as PD-L1-Fc/Oxi-αCD nanoparticles, Fig. 1). The morphology of the Blank/Oxi-αCD nanoparticles and PD-L1-Fc/Oxi-αCD nanoparticles was examined using transmission

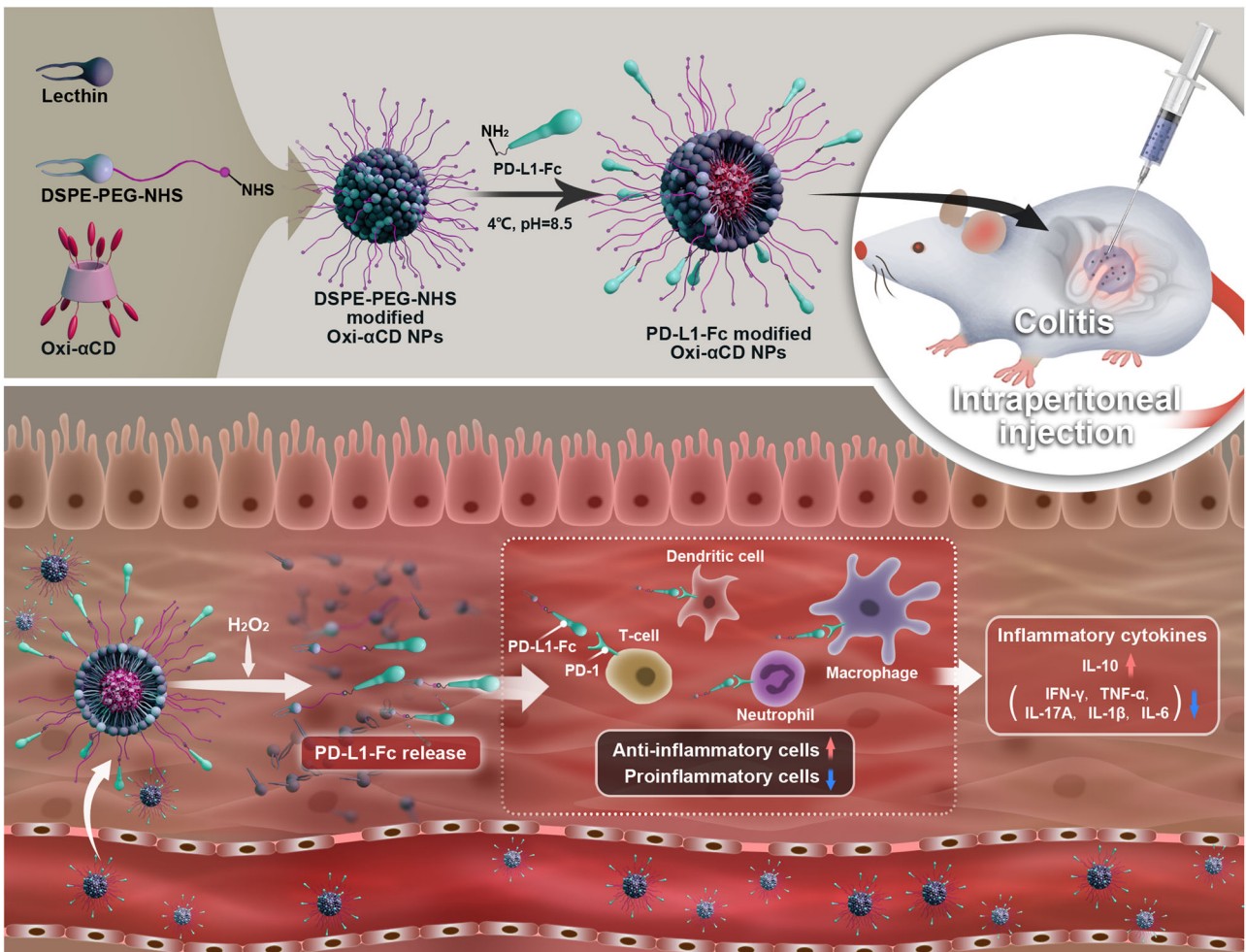

**Fig. 1 | Schematic illustration of engineering of PD-L1-Fc-loaded ROS-responsive nanoparticles for targeted treatment of colitis.** PD-L1-Fc-modified nanoparticles can target the inflamed colon via passing through the blood vessels around the inflammation site, and the loaded PD-L1-Fc is released in response to high levels of ROS in the microenvironment of the diseased colon, which then exerts its multiple biological activities by mediating the frequency of inflammatory cells or inflammatory cytokine production. NPs nanoparticles.

electron microscopy (TEM) and scanning electron microscopy (SEM). Both TEM and SEM results indicated that Blank/Oxi-αCD nanoparticles and PD-L1-Fc/Oxi-αCD nanoparticles have spherical structures (Fig. 2A, B and Supplementary Fig. 1). The hydrodynamic size and surface potential of these nanoparticles were evaluated by dynamic light scattering (DLS) (Fig. 2A, B and Supplementary Table 1). These nanoparticles had a relatively narrow distribution of hydrodynamic size with a low polydispersity index (PDI) of less than 0.2. The mean hydrodynamic size and surface potential of PD-L1-Fc/Oxi-αCD nanoparticles were $221.9 \pm 2.7$ nm and $-18.2 \pm 1.4$ mV, respectively, which were obviously increased compared with those of the Blank/Oxi-αCD nanoparticles ($150.6 \pm 3.2$ nm and $-20.8 \pm 0.5$ mV), indicating that PD-L1-Fc was successfully conjugated to the nanoparticles. An FDA-approved biodegradable material (PLGA) was used as a nonresponsive control material to fabricate PD-L1-Fc-loaded PLGA nanoparticles (defined as PD-L1-Fc/PLGA nanoparticles) by the same method. PD-L1-Fc/Oxi-αCD nanoparticles and PD-L1-Fc/PLGA nanoparticles have comparable physicochemical properties (Supplementary Fig. 3A, B and Supplementary Table 1).

After PD-L1-Fc and Blank/Oxi-αCD nanoparticles had reacted for 48 h, the PD-L1-Fc conjugated on the surface of nanoparticles was calculated by measuring the concentration of free PD-L1-Fc that was separated from the reacted mixture by 100-kDa ultrafiltration devices. The loading ratios of PD-L1-Fc were 45.33% and 58.17% on PD-L1-Fc/Oxi-αCD nanoparticles and PD-L1-Fc/PLGA nanoparticles, respectively. The SDS–PAGE and western blot results showed a consistent molecular weight of protein bands between PD-L1-Fc and purified PD-L1-Fc-loaded nanoparticles, suggesting that PD-L1-Fc was indeed conjugated to the nanoparticles (Fig. 2C, D and Supplementary Fig. 3C, D).

### In vitro hydrolysis of the nanoparticles and release of PD-L1-Fc

To verify the ROS-sensitive and ROS-specific degradability of PD-L1-Fc-loaded nanoparticles, nanoparticles were immersed in ultrapure water with or without 1.0 mM $H_2O_2$ for 6 h. A rapid hydrolysis profile of PD-L1-Fc/Oxi-αCD nanoparticles and Blank/Oxi-αCD nanoparticles showed that the hydrolysis percentage reached approximately 90% within 1 h in an aqueous solution containing 1.0 mM $H_2O_2$, while almost no hydrolysis occurred after 6 h without $H_2O_2$ (Supplementary Fig. 2A). Obvious changes in the morphology of PD-L1-Fc/Oxi-αCD nanoparticles and Blank/Oxi-αCD nanoparticles were observed after 6 h of treatment with 1.0 mM $H_2O_2$ (Supplementary Fig. 2B). A significant decrease in hydrodynamic size with a high PDI (more than 0.4) and a dissociated spatial structure of PD-L1-Fc/Oxi-αCD nanoparticles was visualized by DLS and TEM (Supplementary Fig. 2C, D). However, PD-L1-Fc/PLGA nanoparticles with or without 1.0 mM $H_2O_2$ treatment exhibited very slight hydrolysis and almost remained unchanged in hydrodynamic size and PDI over time (Supplementary Fig. 3E, G). Moreover, the ROS scavenging capability of the nanoparticles was assessed using an $H_2O_2$ assay kit. There was a significant decrease in the residual $H_2O_2$ concentration after incubation with 1 mg/mL PD-L1-Fc/Oxi-αCD nanoparticles but not with PD-L1-Fc/PLGA nanoparticles (Supplementary Fig. 4). These results revealed that PD-L1-Fc/Oxi-αCD nanoparticles had ROS-responsive and ROS-scavenging properties, and the preparation process of loading PD-L1-Fc on Blank/Oxi-αCD nanoparticles preserved the specific responsiveness toward ROS.

Consistently, PD-L1-Fc/Oxi-αCD nanoparticles displayed a rapid release of PD-L1-Fc with 1.0 mM $H_2O_2$ treatment compared to that without $H_2O_2$ treatment. Approximately 90% of the PD-L1-Fc was released within 1 h (Supplementary Fig. 2E). In contrast, with or without 1.0 mM $H_2O_2$ treatment, only approximately 5% of the PD-L1-Fc was released from PD-L1-Fc/PLGA nanoparticles (Supplementary Fig. 3F). To mimic the delivery process in a high ROS environment, Cy5-labeled PD-L1-Fc/Oxi-αCD nanoparticles (containing 20 μg of PD-L1-Fc) were first treated with 0 μM, 100 μM, 200 μM or 500 μM $H_2O_2$ within 1 h, and then the released Cy5-PD-L1-Fc was collected by ultrafiltration. The solution containing Cy5-PD-L1-Fc was incubated with anti-CD3-activated spleen lymphocytes for 8 h. As the $H_2O_2$ concentration increased, more Cy5-PD-L1-Fc was released from the nanoparticles, and the Cy5 fluorescence signal gradually increased on lymphocytes (Supplementary Fig. 2F–H). These results revealed the ROS-responsive drug-release capability of PD-L1-Fc/Oxi-αCD nanoparticles.

### The stability of the nanoparticles

The size and PDI of PD-L1-Fc/Oxi-αCD nanoparticles and PD-L1-Fc/PLGA nanoparticles were not obviously changed after a 24-h incubation with 10% fetal bovine serum (FBS) at 37 °C (Supplementary Fig. 5A, B), indicating that these nanoparticles can maintain a certain stability in serum. Flow cytometry results showed that the PD-L1-Fc-loaded nanoparticles had no differences in cell binding between 0 h and 24 h of 10% FBS incubation (Supplementary Fig. 5C, D), indicating superior serum stability of PD-L1-Fc on these nanoparticles.

### In vitro bioactivity of PD-L1-Fc

To determine the bioactivity of the conjugated or released PD-L1-Fc, PD-L1-Fc binding to lymphocytes and the proliferation inhibition of PD-L1-Fc on lymphocytes were detected by flow cytometry analysis and the CCK-8 assay, respectively. Compared to the IgG1 Fc group, significantly increased fluorescence intensities were found in the PD-L1-Fc/Oxi-αCD nanoparticles, PD-L1-Fc/PLGA nanoparticles, and free PD-L1-Fc groups, indicating that PD-L1-Fc can bind to activated lymphocytes in vitro. Importantly, the released PD-L1-Fc from PD-L1-Fc/Oxi-αCD nanoparticles displayed a similar binding capability to free PD-L1-Fc (Fig. 2E), suggesting that the bioactivity of the released PD-L1-Fc from PD-L1-Fc/Oxi-αCD nanoparticles is preserved. Compared to PD-L1-Fc/Oxi-αCD, PD-L1-Fc/PLGA nanoparticles exhibited less binding to activated lymphocytes with or without $H_2O_2$ treatment (Supplementary Fig. 3H). In addition, the decrease in cellular viability was consistent with the binding capability of PD-L1-Fc to cells (Fig. 2F and Supplementary Fig. 3I), which was associated with the inhibition of lymphocyte proliferation by PD-L1. These results demonstrated that the released PD-L1-Fc from PD-L1-Fc/Oxi-αCD nanoparticles preserved its bioactivity, and PD-L1-Fc/Oxi-αCD showed better bioactivity than PD-L1-Fc/PLGA nanoparticles in an ROS microenvironment.

### Biodistribution of PD-L1-Fc/Oxi-αCD nanoparticles in colitis mice

Colitis in mice was induced by oral administration of 2% DSS in drinking water. To observe the inflammatory site targeting ability of Cy5-PD-L1-Fc and Cy5-PD-L1-Fc/Oxi-αCD nanoparticles, we determined their distributions in the colon of acute colitis mice and healthy mice. At 8 h after intraperitoneal injection, ex vivo imaging of colons revealed that there was greater fluorescence intensity of Cy5-PD-L1-Fc/Oxi-αCD nanoparticles in colons from colitis mice compared to those of healthy mice (Fig. 2G). Compared to free Cy5-PD-L1-Fc, Cy5-PD-L1-Fc/Oxi-αCD nanoparticles exhibited higher fluorescent signals in the diseased colons from 4 h to 48 h, whereas the fluorescent signals of Cy5-PD-L1-Fc reached a maximum at 4 h and almost disappeared after 24 h postinjection (Fig. 2H). The changes in fluorescence intensity were plotted by the fluorescence intensity–time curve and quantified by calculating the area under the fluorescence intensity–time curve (AUC). The AUC of Cy5-PD-L1-Fc/Oxi-αCD nanoparticles was significantly higher than that of Cy5-PD-L1-Fc (Fig. 2I). Consistently, compared to the Cy5-PD-L1-Fc group, Cy5-PD-L1-Fc/Oxi-αCD nanoparticles exhibited lower nonspecific fluorescence distributions in the intestine, liver, spleen and kidneys (Supplementary Fig. 6A–D). These results indicated that the ROS-responsive nanoparticles can effectively enhance PD-L1-Fc accumulation in the inflamed colon and reduce its nonspecific distribution in other organs.

To demonstrate the localization of PD-L1-Fc/Oxi-αCD nanoparticles in the inflamed colon, we prepared PD-L1-Fc/Cy5-Oxi-αCD

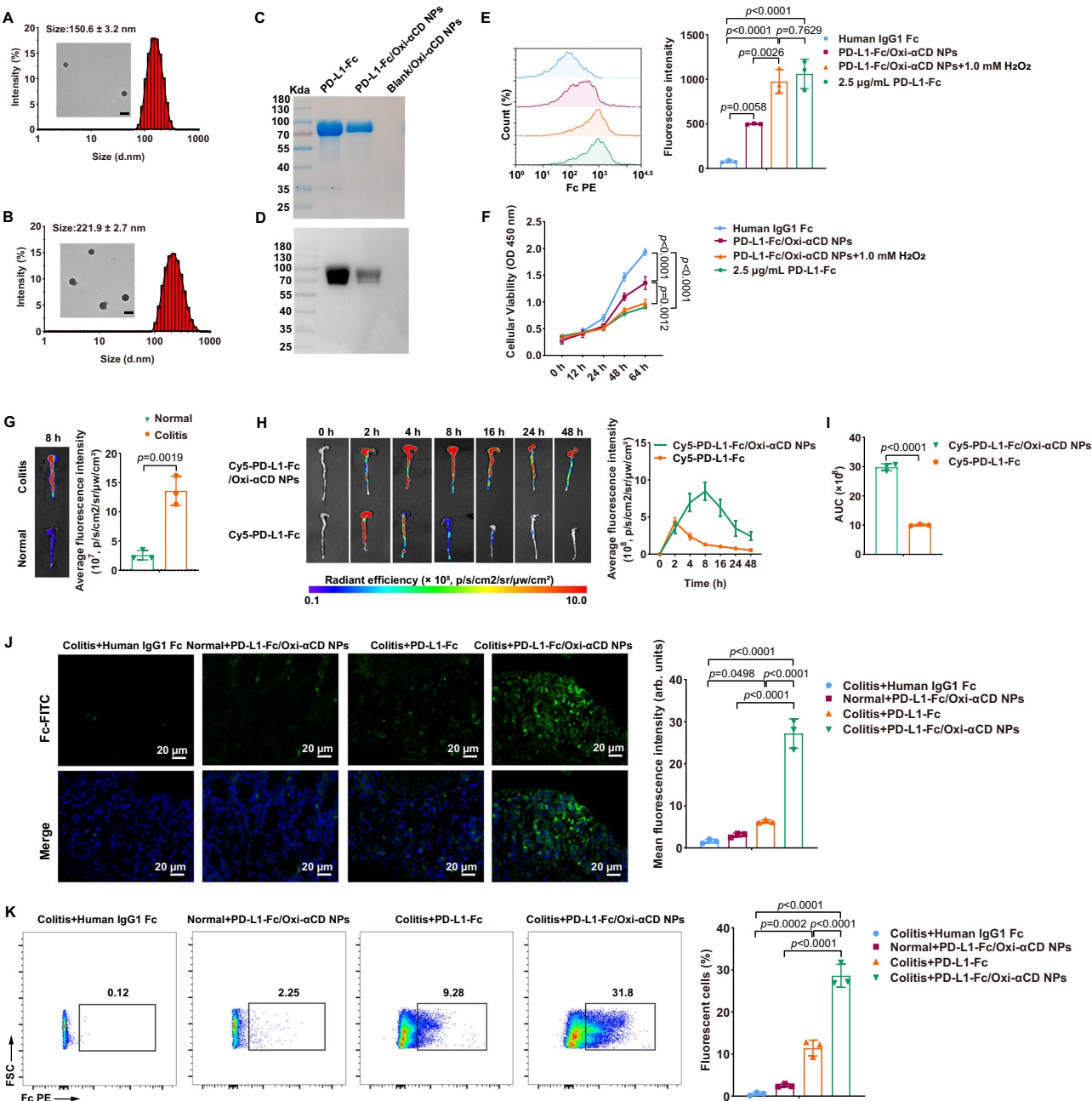

**Fig. 2 | Physiochemical characterization of the PD-L1-Fc/Oxi-αCD nanoparticles and selective accumulation of Cy5-labeled PD-L1-Fc/Oxi-αCD nanoparticles in the colons of DSS-induced mice. A, B** Transmission electron microscopy (TEM) images and size distribution profile of PD-L1-Fc/Oxi-αCD nanoparticles ($n = 3$ independent experiments). Loading of the PD-L1-Fc protein on Blank/Oxi-αCD nanoparticles was confirmed by SDS–PAGE (**C**) and western blotting (**D**). E Flow cytometry analysis of the hydrolyzed and unhydrolyzed PD-L1-Fc/Oxi-αCD nanoparticles binding capability to spleen lymphocytes ($n = 3$ biologically independent samples, $p = 5.75459 \times 10^{-3}$, $3.19572 \times 10^{-5}$, $1.61351 \times 10^{-5}$, $2.64841 \times 10^{-3}$ and $7.62943 \times 10^{-1}$). The fluorescence intensity of PE revealed the bioactivity of PD-L1-Fc after chemical conjugation to the nanoparticles and after release from the nanoparticles. Human IgG1 Fc was used as a negative control. **F** Cell Counting Kit-8 (CCK-8) analysis of the cellular viability of anti-CD3-activated spleen lymphocytes treated with hydrolyzed or unhydrolyzed PD-L1-Fc/Oxi-αCD nanoparticles ($n = 3$ biologically independent samples, $p = 5.91104 \times 10^{-5}$, $1.36050 \times 10^{-6}$ and $1.23415 \times 10^{-3}$). **G** Ex vivo images (left) and quantitative analysis (right) illustrating the distribution

of Cy5-PD-L1-Fc/Oxi-αCD nanoparticles in the colons of mice with or without DSS-induced acute colitis at 8 h after intraperitoneal injection ($n = 3$ mice, $p = 1.87007 \times 10^{-3}$). H Ex vivo images (left) and quantitative data (right) of the colons from colitis mice treated with free Cy5-PD-L1-Fc or Cy5-PD-L1-Fc/Oxi-αCD nanoparticles. **I** The area under the fluorescence intensity-time curve (AUC) (right) of Cy5 fluorescence in the colons after treatment with Cy5-PD-L1-Fc or Cy5-PD-L1-Fc/Oxi-αCD nanoparticles ($n = 3$ mice, $p = 9.37820 \times 10^{-6}$). **J** CLSM images (left) and quantitative data (right) of frozen colonic sections from mice treated with different formulations for approximately 8 h ($n = 3$ mice, $p = 4.98164 \times 10^{-2}$, $5.33800 \times 10^{-7}$, $8.73300 \times 10^{-7}$ and $2.65730 \times 10^{-6}$). Blue channel, nucleus; green channel, PD-L1-Fc. **K** Flow cytometry analysis of the distribution of PD-L1-Fc delivered by different formulations in colonic lamina propria cells from normal or colitis mice ($n = 3$ mice, $p = 2.21326 \times 10^{-4}$, $1.35800 \times 10^{-7}$, $2.65200 \times 10^{-7}$ and $7.46180 \times 10^{-6}$). Data are presented as mean ± SD. P values derived from one-way ANOVA analysis followed by Tukey's multiple comparisons test (**E, F, J, K**) or two-sided Student's $t$ test (**G, I**). Source data are provided as a Source Data file. NPs nanoparticles.

nanoparticles by reacting Cy5-labeled Oxi-αCD nanoparticles with the PD-L1-Fc protein. Healthy mice and colitis mice were treated with PD-L1-Fc/Cy5-Oxi-αCD nanoparticles by intraperitoneal injection for 8 h, and fluorescence imaging of colon cryosections revealed notably accumulated fluorescence in the mucous lamina propria of the colon from colitis mice, not in that from healthy mice (Supplementary Fig. 7). This can be explained by the enhanced vascular permeability in the blood vessels around the inflamed site, which caused nanoparticles to pass through the blood vessels and accumulate in the inflamed colon.

### In vivo targeted delivery of PD-L1-Fc to immune cells

Since the release of PD-L1-Fc from the PD-L1-Fc/Oxi-αCD nanoparticles is dependent on ROS levels in the inflammatory environment, we detected $H_2O_2$ level and PD-L1-Fc location to determine whether PD-L1-Fc was delivered to the inflammatory sites of colonic tissue. We found that $H_2O_2$ level in colonic tissue of colitis mice were significantly higher than those in the colonic tissue of normal mice. At 8 h after intraperitoneal injection of PD-L1-Fc/Oxi-αCD nanoparticles, the $H_2O_2$ concentration in the colonic tissue of the colitis mice decreased significantly, while the $H_2O_2$ concentration remained little changed in the colonic tissue of the normal mice (Supplementary Fig. 8), indicating that the PD-L1-Fc/Oxi-αCD nanoparticles had reached the inflammatory site and reacted with the ROS. In addition, neither PD-L1-Fc nor PD-L1-Fc/PLGA nanoparticles reduced $H_2O_2$ level in colonic tissue of colitis mice.

We then used FITC-conjugated anti-human IgG Fc antibodies (1:1000) to detect PD-L1-Fc protein in frozen colon sections from different treatment groups. At 8 h after intraperitoneal injection of PD-L1-Fc/Oxi-αCD nanoparticles, confocal laser scanning microscopy (CLSM) images revealed that a large amount of FITC fluorescence signal was detected in the colonic mucosa of colitis mice, whereas an extremely weak fluorescence signal was detected in normal mice (Fig. 2J), suggesting that the inflamed environment was beneficial for delivering the PD-L1-Fc protein via PD-L1-Fc/Oxi-αCD nanoparticles. Compared to PD-L1-Fc/Oxi-αCD nanoparticles-treated mice, the colonic mucosa of colitis mice intraperitoneally injected with PD-L1-Fc or PD-L1-Fc/PLGA nanoparticles showed a decreased fluorescence signal (Fig. 2J and Supplementary Fig. 9A). Moreover, we assessed the frequency of PD-L1-Fc-binding positive cells in the colonic lamina propria by flow cytometry. Consistent with the CLSM results, an increased percentage of fluorescence-positive cells was observed in the PD-L1-Fc/Oxi-αCD nanoparticles group of colitis mice 8 h after intraperitoneal injection, which was much higher than that of healthy mice. The percentage of fluorescence-positive cells in the PD-L1-Fc group and PD-L1-Fc/PLGA nanoparticles group was significantly lower than that in the PD-L1-Fc/Oxi-αCD nanoparticles group (Fig. 2K and Supplementary Fig. 9B).

To further demonstrate that PD-L1-Fc/Oxi-αCD nanoparticles target inflammatory cells for PD-L1-Fc delivery, CD4[+] T cells, dendritic cells (DC), macrophages and neutrophils were detected by flow cytometry to assess their binding with the nanoparticles. In the PD-L1-Fc/Oxi-αCD nanoparticles-treated group, these inflammatory cells from colitis mice exhibited significantly higher proportions of fluorescence-positive cells than those from healthy mice (Supplementary Fig. 10A–D). For colitis mice, the percentages of fluorescence-positive cells among these inflammatory cells from the PD-L1-Fc/Oxi-αCD nanoparticles-treated group were significantly higher than those from the PD-L1-Fc-treated group (Supplementary Fig. 10A–D) and PD-L1-Fc/PLGA nanoparticles-treated group (Supplementary Fig. 11A–D). Our results demonstrated that PD-L1-Fc/Oxi-αCD nanoparticles can effectively deliver PD-L1-Fc to inflammatory cells in the colonic lamina propria of mice with enteritis.

### Therapeutic effect of PD-L1-Fc/Oxi-αCD nanoparticles on acute colitis

The therapeutic effect of PD-L1-Fc/Oxi-αCD nanoparticles was evaluated in mice with acute colitis. Compared with the untreated colitis group, the PD-L1-Fc-treated, PD-L1-Fc/PLGA nanoparticles-treated and PD-L1-Fc/Oxi-αCD nanoparticles-treated groups showed less weight loss, lower disease activity index (DAI) scores and longer colon lengths. The above changes in PD-L1-Fc/Oxi-αCD nanoparticles-treated mice were more significant, whereas there was no significant change in Blank/Oxi-αCD nanoparticles-treated mice (Fig. 3B, C, E, F, and Supplementary Fig. 12A–C). PD-L1-Fc/Oxi-αCD nanoparticles effectively alleviated symptoms of rectal bleeding compared with Blank/Oxi-αCD nanoparticles (Fig. 3D). In addition, mini-endoscopic imaging showed a significant improvement in inflammatory appearances in the colonic mucosa of PD-L1-Fc/Oxi-αCD nanoparticles-treated mice compared to that of the free PD-L1-Fc-treated and PD-L1-Fc/PLGA nanoparticles-treated mice (Fig. 3G and Supplementary Fig. 12D).

Histopathological evaluation showed that DSS-induced colitis mice exhibited notable damage to the colon structure with epithelial disruption, loss of goblet cells, and significant infiltration of inflammatory cells, whereas the administration of PD-L1-Fc, PD-L1-Fc/PLGA nanoparticles or PD-L1-Fc/Oxi-αCD nanoparticles alleviated histological colonic damage, as shown by the repaired colonic mucosal epithelium, increased goblet cell frequency, and reduced inflammatory cell infiltration in the mucosa and submucosa (Fig. 3H and Supplementary Fig. 12E). Pathological sections of colons from mice administered PD-L1-Fc/Oxi-αCD nanoparticles showed a nearly normal histological microstructure. The colonic mucosa of mice treated with PD-L1-Fc or PD-L1-Fc/PLGA nanoparticles was also repaired, but the therapeutic effect was inferior to that of PD-L1-Fc/Oxi-αCD nanoparticles.

SEM results showed that a reduced number and irregular spacing of crypts, severely impaired integrity of the epithelial layer, and serious damage to the villi and microvilli on the colonic surface were obviously observed in DSS-induced colitis mice. These pathological manifestations were alleviated with free PD-L1-Fc treatment. Interestingly, these manifestations were significantly improved by PD-L1-Fc/Oxi-αCD nanoparticles treatment compared to the PD-L1-Fc group (Fig. 3I).

To further investigate the anti-inflammatory mechanism of PD-L1-Fc/Oxi-αCD nanoparticles, the levels of inflammatory cytokines in colonic fragments were measured by ELISA. Compared to the levels in the normal group, the proinflammatory cytokines (e.g., IFN-γ, TNF-α, IL-17A, IL-1β and IL-6) were significantly elevated in the colitis group, suggesting that mice bearing colitis have an increase in proinflammatory cytokines. Model mice treated with Blank/Oxi-αCD nanoparticles had lower IFN-γ, TNF-α and IL-6 expression than the model group because they had certain anti-inflammatory activity through the scavenging of ROS[24]. PD-L1-Fc can distinctly decrease proinflammatory factor expression compared to that in model mice, indicating that PD-L1-Fc can serve as an anti-inflammatory therapeutic for colitis treatment. Interestingly, PD-L1-Fc/Oxi-αCD nanoparticles obviously reduced the levels of proinflammatory cytokines and increased the production of the anti-inflammatory cytokine IL-10 compared to the other groups, suggesting that the anti-inflammatory efficacy of PD-L1-Fc was enhanced through nanoparticles delivery. Importantly, the levels of IL-17A and IL-6 showed no significant differences between the PD-L1-Fc/Oxi-αCD nanoparticles group and the normal control group (Fig. 3J), indicating that the inflammation of mice was obviously alleviated through PD-L1-Fc/Oxi-αCD nanoparticles treatment. In conclusion, PD-L1-Fc/Oxi-αCD nanoparticles can obviously relieve inflammation in acute colitis mice through targeted delivery of PD-L1-Fc to inflamed colonic tissues.

### Efficacy of PD-L1-Fc/Oxi-αCD nanoparticles in chronic colitis

To further investigate the efficacy of PD-L1-Fc/Oxi-αCD nanoparticles for chronic colitis, a chronic colitis mouse model was established through repeated challenges with DSS (Fig. 4A). In mice bearing chronic colitis, the body weight of mice was obviously decreased, the DAI increased significantly, and the colon was shortened. These

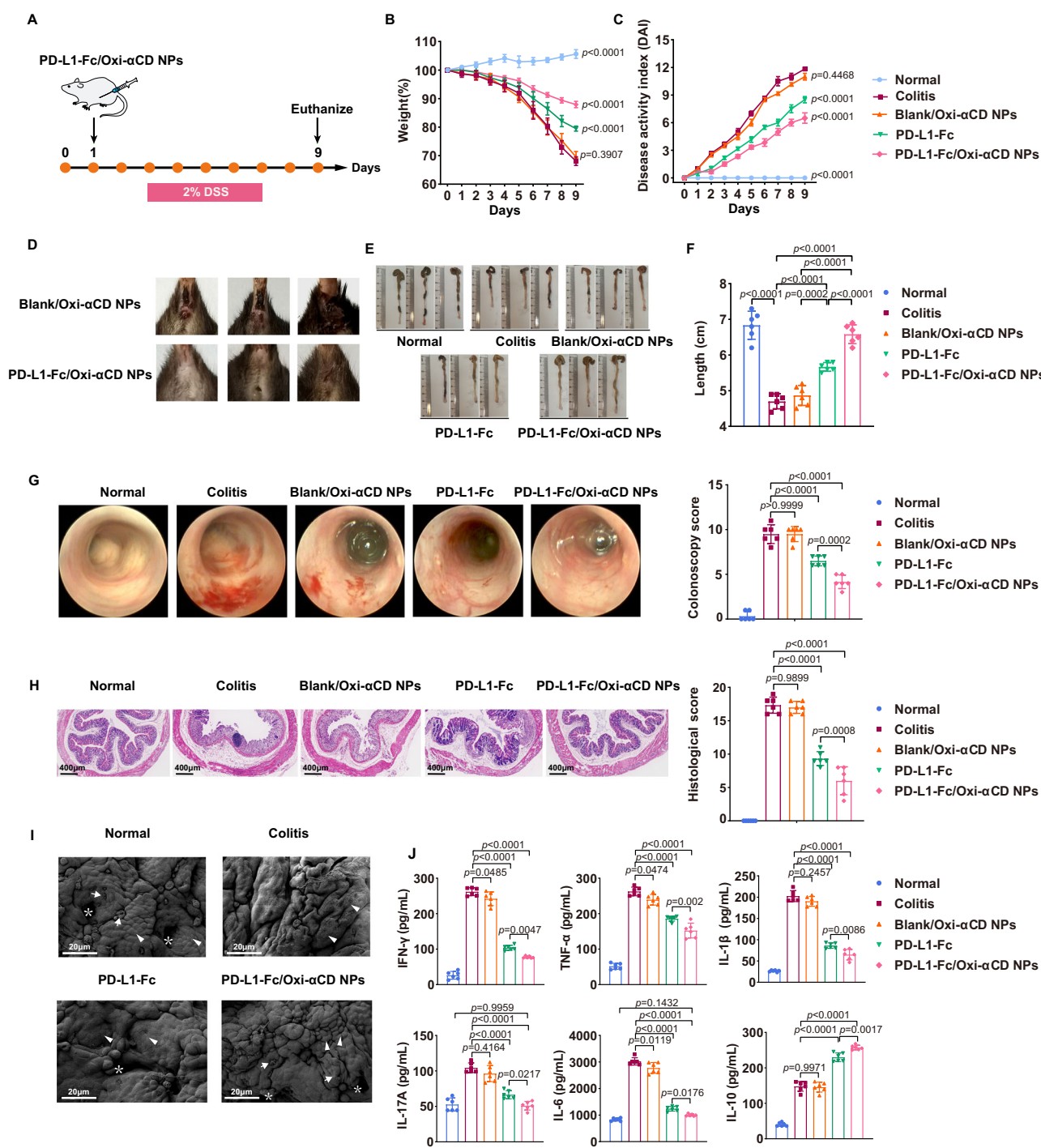

symptoms were improved by PD-L1-Fc, PD-L1-Fc/Oxi-αCD nanoparticles or PD-L1-Fc/PLGA nanoparticles treatment (Fig. 4B–E and Supplementary Fig. 13A–C). Interestingly, PD-L1-Fc/Oxi-αCD nanoparticles led to marked amelioration of weight loss, colonic shortening, and DAI (Fig. 4B–E and Supplementary Fig. 13A–C). Mini-endoscopic imaging and histopathological evaluation showed that signs of colonic mucosal inflammation were observed in colitis mice and alleviated with PD-L1-Fc, PD-L1-Fc/Oxi-αCD nanoparticles or PD-L1-Fc/PLGA nanoparticles treatment (Fig. 4F, G and Supplementary Fig. 13D, E). Importantly, these symptoms almost disappeared with PD-L1-Fc/Oxi-αCD nanoparticles treatment (Fig. 4F, G and Supplementary Fig. 13D, E). The SEM results demonstrated that DSS-induced colonic mucosal damage was significantly ameliorated by PD-L1-Fc, especially PD-L1-Fc/Oxi-αCD nanoparticles treatment (Fig. 4H). In accordance with acute colitis,

proinflammatory factor levels (e.g., IFN-γ, TNF-α, IL-17A, IL-1β and IL-6) were obviously elevated in chronic colitis mice compared to those of normal mice. As expected, PD-L1-Fc/Oxi-αCD nanoparticles strikingly decreased the levels of these proinflammatory factors in colonic tissues and increased the levels of IL-10 compared to the PD-L1-Fc group (Fig. 4I). These results demonstrated that PD-L1-Fc/Oxi-αCD nanoparticles administration was also effective for the treatment of chronic colitis and displayed better efficacy than PD-L1-Fc treatment.

### PD-L1-Fc/Oxi-αCD nanoparticles diminish the frequencies of innate immune cells in the colon of DSS-induced mice

Since DSS-induced colitis is characterized by the infiltration of inflammatory cells into the colon, we analyzed the effect of PD-L1-Fc/Oxi-αCD nanoparticles treatment on innate immune cell recruitment.

**Fig. 3 | Therapeutic effect of PD-L1-Fc versus PD-L1-Fc/Oxi-αCD nanoparticles on acute DSS-induced colitis. A** Schematic diagram of colitis model preparation and treatment protocol. Changes in body weight (**B**) and DAI scores (**C**) of mice in each group during 9 days of treatment. Comparisons were performed between the colitis group and other groups. In panel (**B**), $n = 6$ mice, $p = 3.30000 \times 10^{-14}$, $3.90705 \times 10^{-1}$, $2.92800 \times 10^{-12}$ and $3.30000 \times 10^{-14}$. In (**C**), $n = 6$ mice, $p = 3.30000 \times 10^{-14}$, $4.46794 \times 10^{-1}$, $3.41556 \times 10^{-6}$ and $4.92972 \times 10^{-10}$. **D** PD-L1-Fc/Oxi-αCD nanoparticles effectively alleviated symptoms of rectal bleeding in mice with enteritis compared with treatment with Blank/Oxi-αCD nanoparticles. Representative digital photos (**E**) and quantified lengths (**F**) of colonic tissues isolated from mice after treatment ($n = 6$ mice, $p = 3.65600 \times 10^{-12}$, $1.51912 \times 10^{-5}$, $5.55390 \times 10^{-11}$, $2.25584 \times 10^{-4}$, $3.94186 \times 10^{-10}$ and $3.38584 \times 10^{-5}$). **G** Representative mini-endoscopic images of colons from mice in the different treatment groups ($n = 6$ mice, $p = 3.87125 \times 10^{-6}$, $9.99900 \times 10^{-1}$, $6.26880 \times 10^{-11}$ and $1.64522 \times 10^{-4}$). The right panel indicates the quantification of the severity of DSS-induced acute colitis. **H** H&E-stained histological sections of colons ($n = 6$ mice, $p = 3.35138 \times 10^{-10}$, $9.89870 \times 10^{-1}$, $1.94000 \times 10^{-13}$ and $8.22848 \times 10^{-4}$). The right panel indicates the pathology score of colons from mice in the different treatment groups. **I** Scanning electron microscopy histological images of colon tissue in each group. DSS-induced acute colitis resulted in loss of goblet cells (white arrows) and intestinal glands (white asterisks) and damaged epithelial cells (white triangles). Treatment with PD-L1-Fc or PD-L1-Fc/Oxi-αCD nanoparticles rescued the loss of goblet cells and intestinal glands and resulted in intact epithelium. **J** The levels of important inflammatory cytokines (IFN-γ, TNF-α, IL-17A, IL-1β, IL-6 and IL-10) in colonic tissues isolated from healthy or colitis mice treated with different formulations ($n = 6$ mice). For IFN-γ, $p = 3.30000 \times 10^{-14}$, $4.84904 \times 10^{-2}$, $3.30000 \times 10^{-14}$ and $4.67598 \times 10^{-3}$; For TNF-α, $p = 5.14678 \times 10^{-9}$, $4.73548 \times 10^{-2}$, $2.28100 \times 10^{-12}$ and $2.02060 \times 10^{-3}$; For IL-1β, $p = 3.30000 \times 10^{-14}$, $2.45742 \times 10^{-1}$, $3.30000 \times 10^{-14}$ and $8.59620 \times 10^{-3}$; For IL-17A, $p = 1.69100 \times 10^{-7}$, $4.16393 \times 10^{-1}$, $1.94936 \times 10^{-10}$, $2.16543 \times 10^{-2}$ and $9.95935 \times 10^{-1}$; For IL-6, $p = 3.30000 \times 10^{-14}$, $1.18992 \times 10^{-2}$, $3.30000 \times 10^{-14}$, $1.76390 \times 10^{-2}$ and $1.43236 \times 10^{-1}$; For IL-10, $p = 1.17420 \times 10^{-11}$, $9.97135 \times 10^{-1}$, $5.10000 \times 10^{-14}$ and $1.68021 \times 10^{-3}$. Data are presented as mean ± SD. *P* values derived from One-way ANOVA analysis followed by Tukey's multiple comparisons test (**B, C, F**, right panel of **G**, right panel of **H, J**). Source data are provided as a Source Data file. NPs nanoparticles.

Flow cytometry was employed to analyze the frequencies of DCs (MHC-II⁺ CD11c⁺), macrophages (CD11b⁺ F4/80⁺) and neutrophils (CD11b⁺ LY6G⁺) in colonic lamina propria (LP) tissues isolated from healthy mice and DSS-induced acute or chronic colitis mice. Flow cytometry analysis showed decreased frequencies of DCs, macrophages and neutrophils in colitis mice treated with soluble PD-L1-Fc, PD-L1-Fc/PLGA nanoparticles or PD-L1-Fc/Oxi-αCD nanoparticles compared with untreated colitis mice or colitis mice treated with Blank/Oxi-αCD nanoparticles, with more obvious decreases in the frequencies of these cells in PD-L1-Fc/Oxi-αCD nanoparticles-treated colitis mice (Figs. 5A–C, 6A–C, Supplementary Fig. 14A–C and 15A–C). These results suggest that PD-L1-Fc can inhibit the recruitment of innate immune cells into the colon during DSS-induced colitis and that the inhibitory effect of PD-L1-Fc/Oxi-αCD nanoparticles is stronger than that of free PD-L1-Fc and PD-L1-Fc/PLGA nanoparticles.

### PD-L1-Fc/Oxi-αCD nanoparticles increase the frequencies of adaptive immune cells in the colon of DSS-induced mice

In addition to innate immune cells, some adaptive immune cells can affect the progression of colitis. A previous study suggested that PD-L1-Fc could significantly reduce the severity of colitis by inhibiting T helper 17 (Th17) responses but promoting Th1 responses during the development of DSS-induced colitis[17]. Tregs and Tfh cells have been reported to alleviate the symptoms of colitis[25,26]. We determined the frequencies of Tregs (CD4⁺FOXP3⁺), Th1 cells (CD4⁺CXCR3⁺), and Tfh cells (CD4⁺CXCR5⁺) in the colonic lamina propria in DSS-induced colitis. Flow cytometry results showed that DSS-induced colitis reduced the frequencies of these cells among colonic lamina propria cells, and soluble PD-L1-Fc, PD-L1-Fc/PLGA nanoparticles or PD-L1-Fc/Oxi-αCD nanoparticles significantly increased the Treg (Figs. 5D, 6D, Supplementary Figs. 14D and 15D), Th1-cell (Figs. 5E, 6E, Supplementary Figs. 14E and 15E), and Tfh-cell frequencies in the colonic lamina propria of colitis mice (Figs. 5F, 6F, Supplementary Figs. 14F and 15F). Compared with PD-L1-Fc-treated and PD-L1-Fc/PLGA nanoparticles-treated colitis mice, PD-L1-Fc/Oxi-αCD nanoparticles-treated colitis mice exhibited a more significant increase in the frequencies of these cells.

### Preliminary in vivo safety evaluation of PD-L1-Fc/Oxi-αCD nanoparticles

The systemic toxicity of PD-L1-Fc restricts its in vivo application. To evaluate whether nanoparticles delivery can decrease the systemic toxicity of PD-L1-Fc, we determined the preliminary in vivo safety of PD-L1-Fc and PD-L1-Fc/Oxi-αCD nanoparticles. H&E staining of major organs from mice with chronic enteritis showed that PD-L1-Fc/Oxi-αCD nanoparticles did not cause significant histopathological changes, whereas PD-L1-Fc caused a decrease in the white pulp of the spleen. These results suggested that PD-L1-Fc caused damage to immune organs, but PD-L1-Fc/Oxi-αCD nanoparticles did not (Supplementary Figs. 16 and 17). Next, the potential side effects after intraperitoneal injection of a high dose of PD-L1-Fc/Oxi-αCD nanoparticles (5.0 g/kg) were examined on the 12th day. The H&E-stained results of gastrointestinal tissues and major organs showed no discernable injuries between the saline group and PD-L1-Fc/Oxi-αCD nanoparticles group (Supplementary Fig. 18A, B), suggesting that PD-L1-Fc/Oxi-αCD nanoparticles displayed safety for intraperitoneal injection.

To determine whether PD-L1-Fc and PD-L1-Fc/Oxi-αCD nanoparticles have side effects on immunity, healthy female C57BL/6J mice were treated with cyclophosphamide (CTX) on days 1, 3 and 5 and then intraperitoneally injected with saline, 40 μg of PD-L1-Fc or PD-L1-Fc/Oxi-αCD nanoparticles (containing 40 μg PD-L1-Fc). Neither PD-L1-Fc nor PD-L1-Fc/Oxi-αCD nanoparticles changed the thymus or spleen size in healthy mice. PD-L1-Fc significantly reduced the size of the thymus but not that of the spleen in CTX-treated mice, whereas PD-L1-Fc/Oxi-αCD nanoparticles did not change the size of the thymus or spleen (Fig. 7A, B). Compared to the saline group, both the CTX-treated and PD-L1-Fc-treated groups had obviously decreased amounts of white blood cells, neutrophils, lymphocytes and monocytes, but the PD-L1-Fc/Oxi-αCD nanoparticles-treated group did not (Fig. 7D–G). There was no significant change in body weight in the mice treated with any of the treatments (Fig. 7H). CTX resulted in a decrease in the thymus index but no change in the spleen index, and injection of PD-L1-Fc or PD-L1-Fc/Oxi-αCD nanoparticles had no significant effect on the organ indexes (Fig. 7I, J).

Furthermore, we constructed a pulmonary infection model by injecting *Pseudomonas aeruginosa* into the nasal cavity of mice treated with saline, PD-L1-Fc or PD-L1-Fc/Oxi-αCD nanoparticles. Compared with the infected saline-treated group, the infected PD-L1-Fc-treated group showed a destroyed, incomplete alveolar wall structure and more erythrocytes in the alveolar space in the lungs, whereas the lungs of PD-L1-Fc/Oxi-αCD nanoparticles-treated mice infected with bacteria were similar to those of mice in the saline group (Fig. 7C). The above results suggested that PD-L1-Fc/Oxi-αCD nanoparticles did not produce an immunosuppressive effect similar to that of PD-L1-Fc and would not cause side effects.

### Regulation of PD-L1-Fc/Oxi-αCD nanoparticles on the DSS-induced colitis gut microbiota

To investigate whether PD-L1-Fc/Oxi-αCD nanoparticles treatment could change the gut microbiota of colitis mice, the gut microbiota in the feces of mice was analyzed. The 16S rRNA gene sequencing results

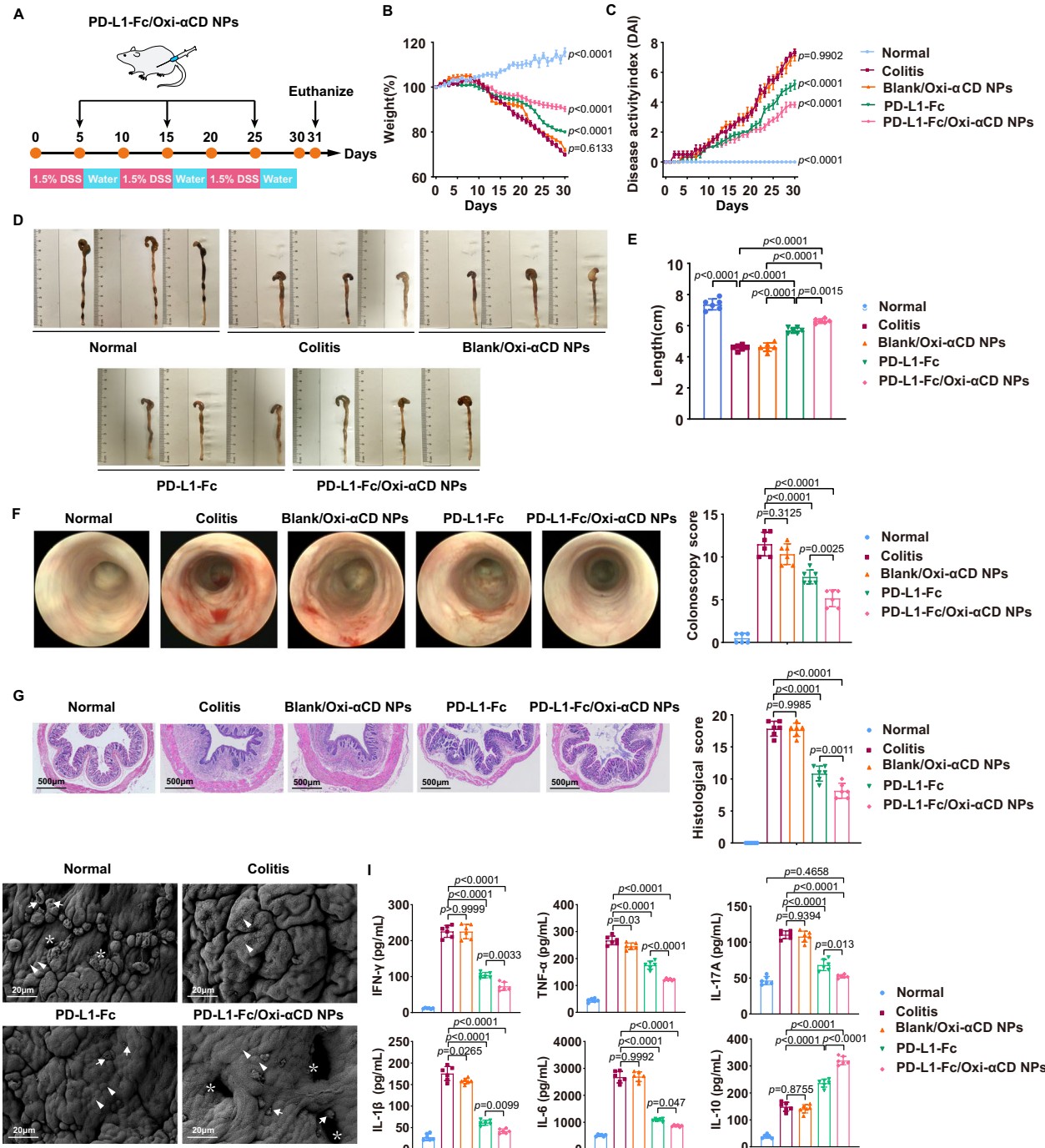

**Fig. 4 | Effects of PD-L1-Fc versus PD-L1-Fc/Oxi-αCD nanoparticles treatment on the development of DSS-induced chronic colitis. A** Preparation of the chronic colitis model and the protocol for the treatment procedure. Changes in body weight (**B**) and DAI scores (**C**) of mice in each group during 32 days of treatment. Comparisons were performed between the colitis group and other groups. In (**B**), $n = 6$ mice, $p = 3.30000 \times 10^{-14}$, $6.13254 \times 10^{-1}$, $4.57936 \times 10^{-6}$ and $3.95500 \times 10^{-12}$. In (**C**), $n = 6$ mice, $p = 3.30000 \times 10^{-14}$, $9.90192 \times 10^{-1}$, $2.77006 \times 10^{-5}$ and $6.09039 \times 10^{-9}$. Representative digital photos (**D**), quantified lengths (**E**), and mini-endoscopic images (**F**) of colons from mice in the different treatment groups. In (**E**), $n = 6$ mice, $p = 3.30000 \times 10^{-14}$, $1.41468 \times 10^{-7}$, $2.40040 \times 10^{-11}$, $2.45988 \times 10^{-7}$, $3.66850 \times 10^{-11}$ and $1.49054 \times 10^{-3}$. In (**F**), $n = 6$ mice, $p = 8.75812 \times 10^{-6}$, $3.12454 \times 10^{-1}$, $8.47678 \times 10^{-10}$ and $2.47231 \times 10^{-3}$. **G** H&E-stained histological sections of colons ($n = 6$ mice, $p = 7.94880 \times 10^{-11}$, $9.98476 \times 10^{-1}$, $9.50000 \times 10^{-14}$ and $1.05638 \times 10^{-3}$). **H** Scanning electron microscopy histological images of colon tissue

in each group. DSS-induced acute colitis resulted in loss of goblet cells (white arrows) and intestinal glands (white asterisks) and damaged epithelial cells (white triangles). **I** The levels of inflammatory cytokines (IFN-γ, TNF-α, IL-17A, IL-1β, IL-6 and IL-10) in colonic tissues isolated from healthy or colitis mice treated with different treatments ($n = 6$ mice). For IFN-γ, $p = 1.57000 \times 10^{-13}$, $9.99993 \times 10^{-1}$, $3.30000 \times 10^{-14}$ and $3.27647 \times 10^{-3}$; For TNF-α, $p = 6.02600 \times 10^{-12}$, $3.00000 \times 10^{-2}$, $3.30000 \times 10^{-14}$ and $2.77366 \times 10^{-7}$; For IL-1β, $p = 3.30000 \times 10^{-14}$, $2.65410 \times 10^{-2}$, $3.30000 \times 10^{-14}$ and $9.88879 \times 10^{-3}$; For IL-17A, $p = 7.69270 \times 10^{-11}$, $9.39431 \times 10^{-13}$, $1.01000 \times 10^{-13}$, $1.30598 \times 10^{-3}$ and $4.65849 \times 10^{-1}$; For IL-6, $p = 3.30000 \times 10^{-14}$, $9.99194 \times 10^{-1}$, $3.30000 \times 10^{-14}$ and $4.70035 \times 10^{-2}$; For IL-10, $p = 7.08200 \times 10^{-10}$, $8.75465 \times 10^{-1}$, $3.30000 \times 10^{-14}$ and $1.90050 \times 10^{-9}$. Data are presented as mean ± SD. $P$ values derived from One-way ANOVA analysis followed by Tukey's multiple comparisons test (**B**, **C**, **E**, right **F**, right panel of **G**, **I**). Source data are provided as a Source Data file. NPs nanoparticles.

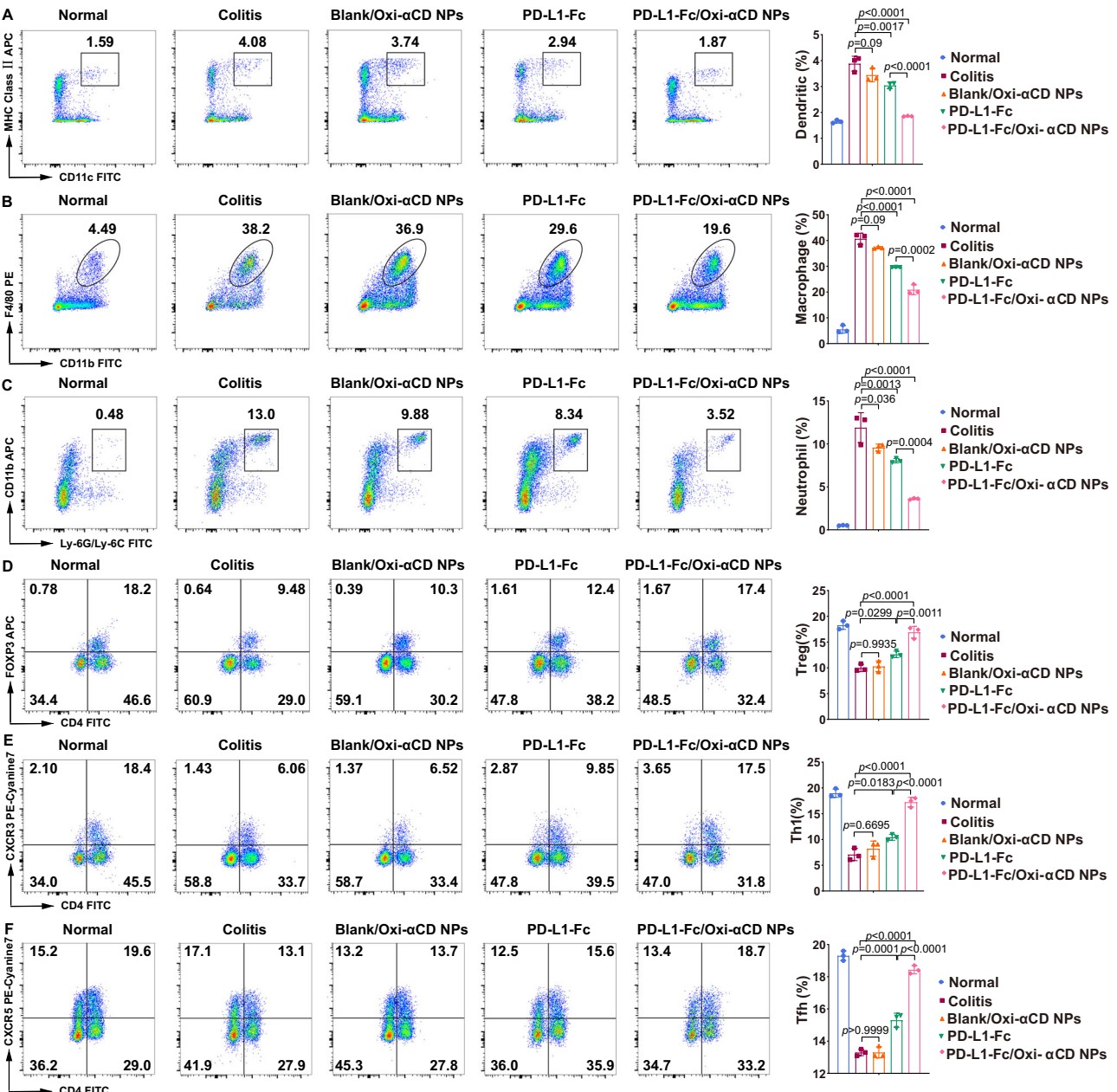

**Fig. 5 | The regulatory effect of PD-L1-Fc versus PD-L1-Fc/Oxi-αCD nanoparticles on the frequency of inflammatory cells derived from colonic tissues of mice with acute colitis.** Five days post-DSS administration, the frequencies of dendritic (**A**), macrophage (**B**), neutrophil (**C**), Treg (**D**), Th1 (**E**), and Tfh (**F**) cells in isolated colonic lamina propria (LP) cells from the indicated groups were determined by flow cytometry. In (**A**), $p = 1.70043 \times 10^{-3}$, $8.99799 \times 10^{-2}$, $7.42287 \times 10^{-7}$ and $9.63563 \times 10^{-5}$. In (**B**), $p = 2.45792 \times 10^{-5}$, $9.00439 \times 10^{-2}$, $1.10640 \times 10^{-7}$ and $1.88860 \times 10^{-4}$. In (**C**), $p = 1.26559 \times 10^{-3}$, $3.60459 \times 10^{-2}$, $1.54738 \times 10^{-6}$ and $3.98454 \times 10^{-4}$. In (**D**), $p = 2.99412 \times 10^{-2}$, $9.93452 \times 10^{-1}$, $1.86512 \times 10^{-5}$ and $1.06227 \times 10^{-3}$. In (**E**), $p = 1.83431 \times 10^{-2}$, $6.69535 \times 10^{-1}$, $2.33772 \times 10^{-6}$ and $8.71996 \times 10^{-5}$. In (**F**), $p = 1.21551 \times 10^{-4}$, $1.00000$, $1.90017 \times 10^{-8}$ and $2.10744 \times 10^{-6}$. All data are presented as mean ± SD ($n = 3$ mice). $P$ values derived from One-way ANOVA analysis followed by Tukey's multiple comparisons test. Source data are provided as a Source Data file. NPs nanoparticles.

showed a significantly increased proportion of *Escherichia−Shigella* and a moderately decreased proportion of *Prevotellaceae* in the feces of DSS-induced mice compared with those of normal mice (Fig. 8A). *Escherichia−Shigella* are well-known infectious pathogens that contribute to the development of colitis, and *Prevotellaceae* can use nondigestible carbohydrates in the intestines to produce short-chain fatty acids (SCFAs) as energy substrates and exert anti-inflammatory effects[27,28]. PD-L1-Fc/Oxi-αCD nanoparticles treatment increased *Prevotellaceae* levels and reduced the expansion of *Escherichia−Shigella* (Fig. 8A). Principal coordinate analysis (PCoA) was employed to evaluate beta diversity, and the difference in community structures between normal mice and colitis mice with or without treatment was

revealed by an apparent clustering separation between amplicon sequence variants (ASVs) (Fig. 8B). The microbial community richness indicated by the Ace estimator showed significant changes among the five groups (Fig. 8C). Kyoto Encyclopedia of Genes and Genomes (KEGG) analysis showed that the target genes of the colonic microbiota from different groups were enriched in metabolic pathways (Fig. 8D). Microorganisms and their metabolites, such as SCFAs, which have been demonstrated to be closely associated with reducing the risk of IBD, play important roles in maintaining intestinal homeostasis. We determined the levels of seven SCFAs in the feces of mice with different treatments, and the results showed that PD-L1-Fc/Oxi-αCD nanoparticles administration could reverse the decrease in SCFA levels

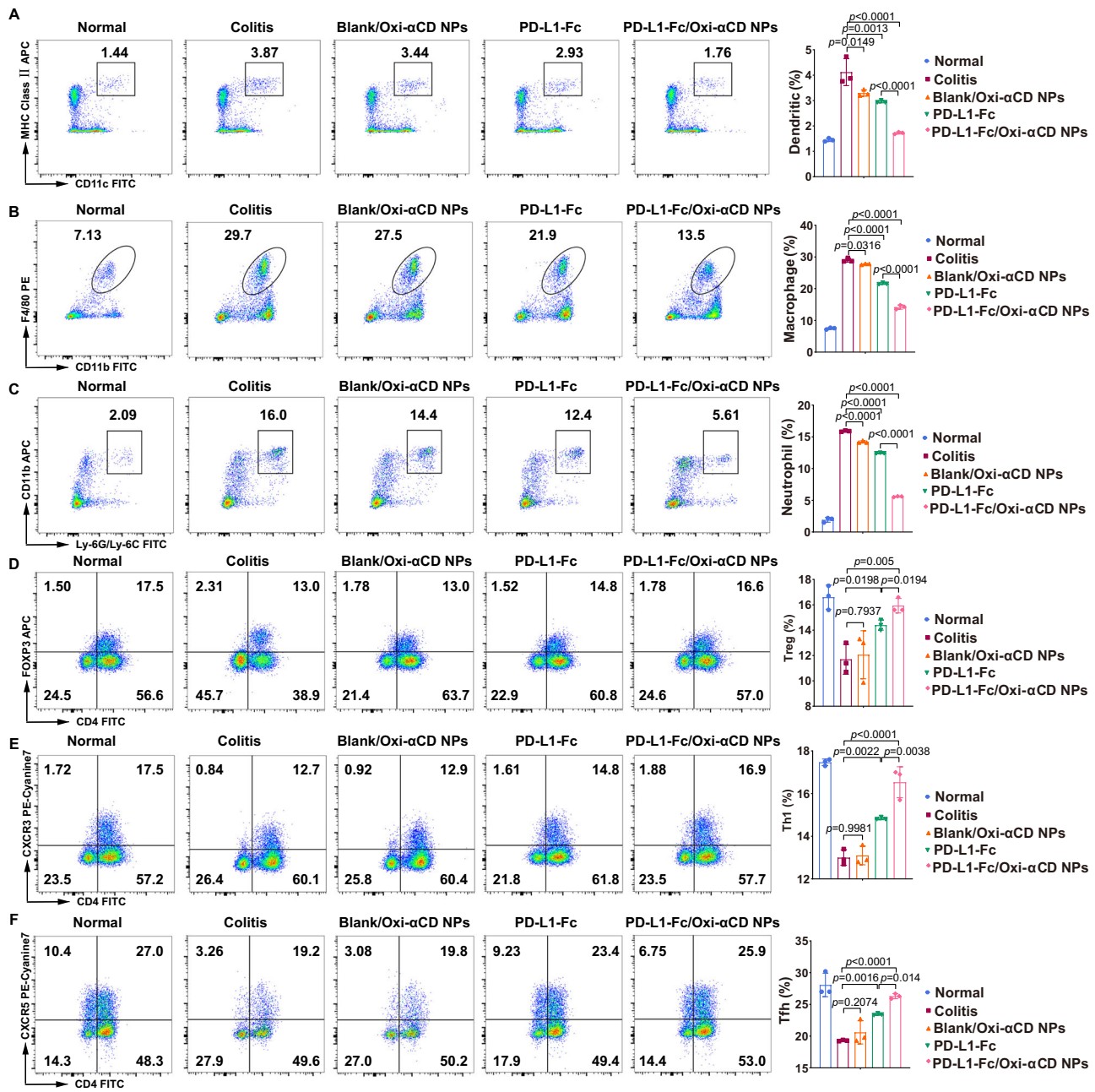

**Fig. 6 | The regulatory effect of PD-L1-Fc/Oxi-αCD nanoparticles on the frequency of inflammatory cells in the colonic tissues of mice with chronic colitis.** After three cycles of DSS administration, the frequencies of dendritic (**A**), macrophage (**B**), neutrophil (**C**), Treg (**D**), Th1 (**E**), and Tfh (**F**) cells in isolated colonic lamina propria (LP) cells from the indicated groups were determined by flow cytometry. In (**A**), $p = 1.33243 \times 10^{-3}$, $1.49082 \times 10^{-2}$, $2.53180 \times 10^{-6}$ and $8.45033 \times 10^{-4}$. In (**B**), $p = 1.35429 \times 10^{-8}$, $3.15662 \times 10^{-2}$, $3.34360 \times 10^{-11}$ and

$1.35429 \times 10^{-8}$. In (**C**), $p = 6.35071 \times 10^{-9}$, $6.79442 \times 10^{-6}$, $6.80000 \times 10^{-14}$ and $2.05430 \times 10^{-11}$. In (**D**), $p = 1.98386 \times 10^{-2}$, $7.93715 \times 10^{-1}$, $5.03937 \times 10^{-3}$ and $1.94161 \times 10^{-2}$. In (**E**), $p = 2.21669 \times 10^{-3}$, $9.98089 \times 10^{-1}$, $8.78094 \times 10^{-6}$ and $3.84292 \times 10^{-3}$. In (**F**), $p = 1.59735 \times 10^{-3}$, $2.07438 \times 10^{-1}$, $2.73335 \times 10^{-5}$ and $1.39921 \times 10^{-2}$. Data are presented as mean ± SD ($n = 3$ mice). $P$ values derived from One-way ANOVA analysis followed by Tukey's multiple comparisons test. Source data are provided as a Source Data file. NPs nanoparticles.

caused by DSS-induced colitis (Fig. 8E). Taken together, our data suggested that PD-L1-Fc/Oxi-αCD nanoparticles can be conductive to improve the bowel environment though shaping the gut microbiota and increasing SCFA levels.

## Discussion

Inflammatory bowel disease (IBD) is a chronic, progressive, relapsing and remitting disorder caused by the interplay among genetic susceptibility, environmental factors, intestinal barrier and immune system dysregulation[1]. Currently, the drugs for IBD therapy include anti-inflammatory agents, antibiotics, immunosuppressers and biological

agents. Although some classic treatment strategies for IBD conquered peculiar roles in the overall IBD strategy, these therapies generally cause significant side effects[29,30]. Biological therapies for IBD have been developed to target proinflammatory and anti-inflammatory cytokines, such as anti-TNF-α agents, revolutionizing IBD treatment[31,32]. However, there have been concerns about antibody responses, leaving IBD patients open to infection with biological therapies. Therefore, a therapeutic agent with enhanced safety and minimal toxicity is necessary for the treatment of IBD.

PD-L1 expressed in intestinal tissue functions as an essential ligand for innate immune cells to prevent gut inflammation[15]. Recombinant

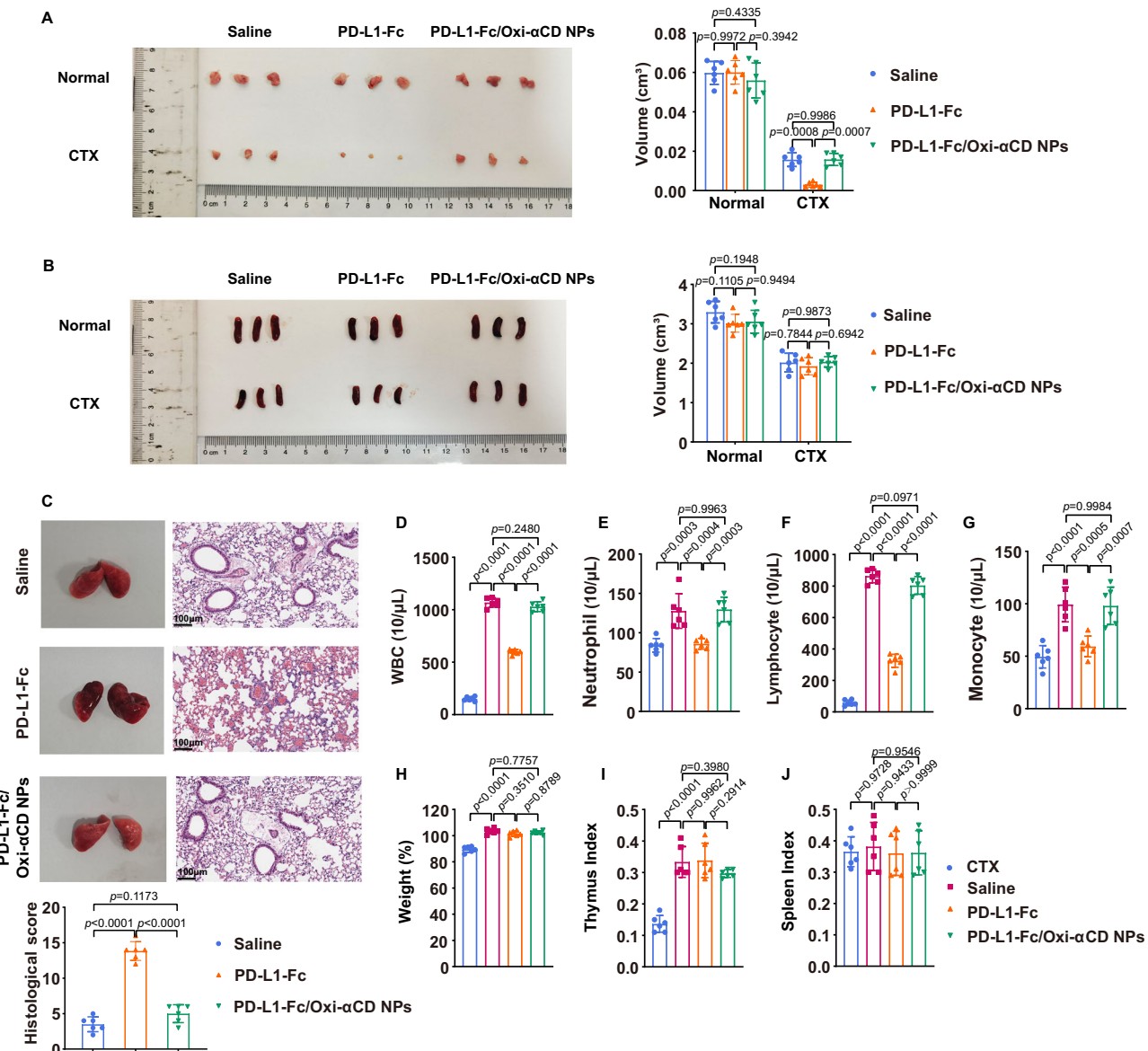

**Fig. 7 | Evaluation of the side effects of PD-L1-Fc/Oxi-αCD nanoparticles in vivo.**
**A**, **B** Images of the spleen and thymus from mice treated with the indicated groups. The right panels indicate the volumes of the spleen and thymus from mice in different treatment groups. In (**A**), $p = 9.97186 \times 10^{-1}$, $4.33501 \times 10^{-1}$, $3.94221 \times 10^{-1}$, $8.22806 \times 10^{-4}$, $9.98648 \times 10^{-1}$ and $7.18600 \times 10^{-4}$. In (**B**), $p = 1.10451 \times 10^{-1}$, $1.94779 \times 10^{-1}$ and $9.49450 \times 10^{-1}$, $7.84366 \times 10^{-1}$, $9.87280 \times 10^{-1}$ and $6.94237 \times 10^{-1}$. **C** Images and H&E-stained sections of the lung tissues from normal and infected mice with the indicated treatment. The lungs from infected mice showed diffuse alveolar damage (blue arrows), and the infected mice treated with PD-L1-Fc had more alveolar hemorrhage than the infected mice treated with saline and PD-L1-Fc/Oxi-αCD nanoparticles. The histological score of the lung tissue indicates the quantification of the severity of infected lung tissues from mice in different treatment groups ($p = 7.58464 \times 10^{-10}$, $1.17331 \times 10^{-1}$ and

$6.83214 \times 10^{-9}$). Routine blood test (**D**–**G**), body weight (**H**) and organ index (**I**–**J**) of mice in the indicated groups. In (**D**), $p = 2.30000 \times 10^{-14}$, $2.80000 \times 10^{-14}$, $2.48015 \times 10^{-1}$ and $4.80000 \times 10^{-14}$. In (**E**), $p = 2.55389 \times 10^{-4}$, $4.36049 \times 10^{-4}$, $9.96278 \times 10^{-1}$ and $2.66995 \times 10^{-4}$. In (**F**), $p = 2.30000 \times 10^{-14}$, $3.10000 \times 10^{-14}$, $9.71271 \times 10^{-2}$ and $9.40000 \times 10^{-14}$. In (**G**), $p = 3.32176 \times 10^{-5}$, $4.80545 \times 10^{-4}$, $9.98410 \times 10^{-1}$ and $6.95958 \times 10^{-4}$. In (**H**), $p = 5.66120 \times 10^{-10}$, $3.50953 \times 10^{-1}$, $7.75738 \times 10^{-1}$ and $8.78878 \times 10^{-1}$. In (**I**), $p = 2.08209 \times 10^{-7}$, $9.96169 \times 10^{-1}$, $3.97959 \times 10^{-1}$ and $2.91408 \times 10^{-1}$. In (**J**), $p = 9.72792 \times 10^{-1}$, $9.43307 \times 10^{-1}$, $9.54551 \times 10^{-1}$ and $9.99971 \times 10^{-1}$. All data are presented as mean ± SD ($n = 6$ mice). $P$ values derived from two-way ANOVA analysis followed by Tukey's multiple comparisons test (right panel of **A**, **B**), or one-way ANOVA analysis followed by Tukey's multiple comparisons test (bottom panel of **C**, **D**–**J**). Source data are provided as a Source Data file. NPs nanoparticles.

Fc-conjugated PD-L1 is a promising biologic for achieving anti-inflammatory effects with IBD therapy due to the suppression of immune cells and their cytokine production via interactions with PD-1 and B7-1[17]. However, engagement of PD-L1 by its receptor may be associated with a decreased ability of the immune system to fight off infection[33]. In our study, spleens from DSS-induced colitis mice treated with PD-L1-Fc showed a decrease in white pulp, and the lungs of PD-L1-Fc-treated mice were more susceptible to *Pseudomonas aeruginosa* infection; other side effects were also present in PD-L1-Fc-treated mice.

This safety problem of PD-L1-Fc limits its further use in IBD therapy; therefore, developing a method of targeted delivery of PD-L1-Fc to the inflamed area in IBD is a strategy to address the side effects of PD-L1-Fc.

Nanotechnology can be used to deliver medications directly to the area of inflammation without exposure to the remaining body, thus avoiding systemic drug-associated side effects[34–37]. Active inflammation in IBD is directly coupled to the generation and release of ROS by immune cells[38]. We and other research groups previously confirmed that nanoparticles based on a ROS-responsive material

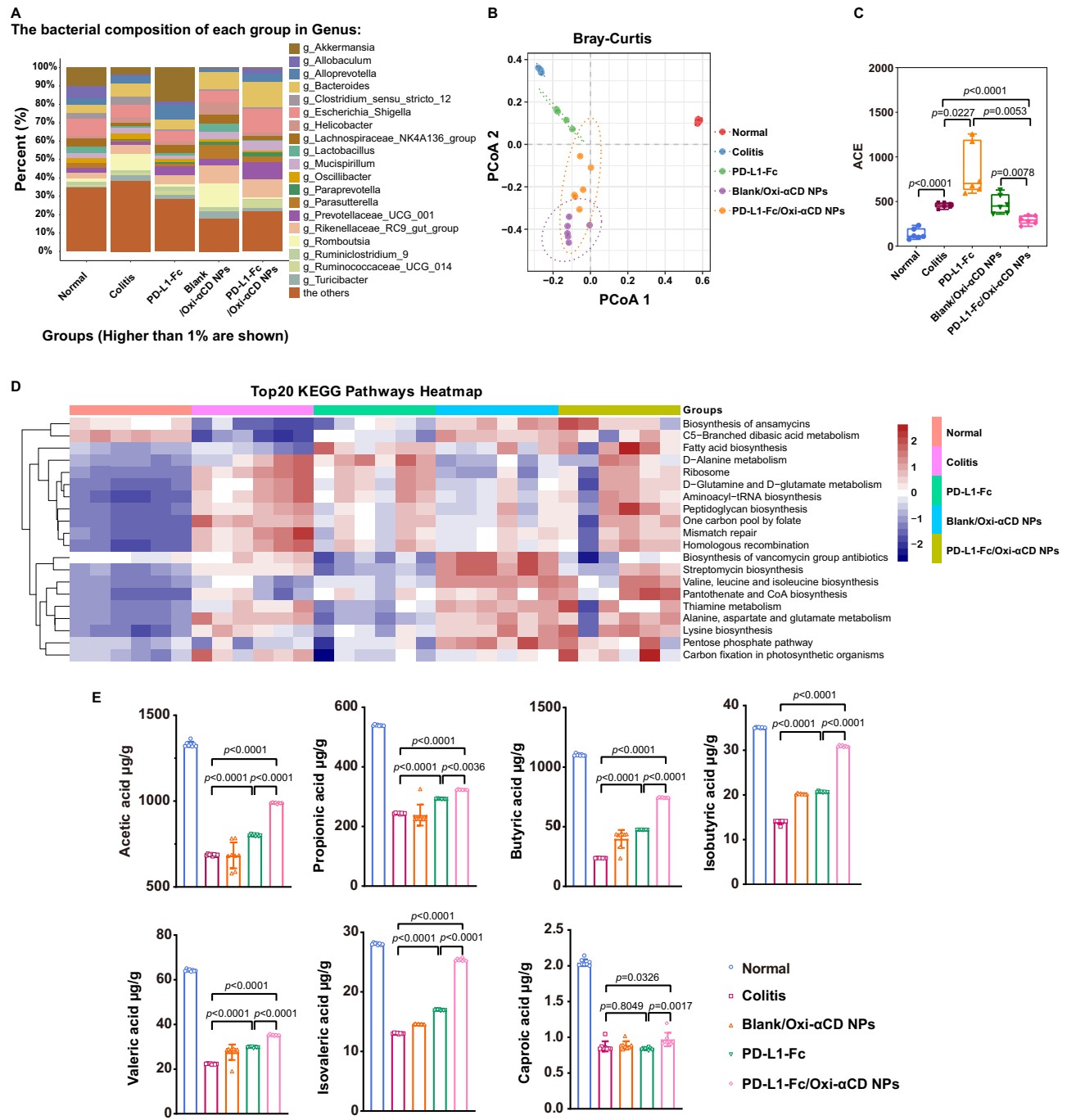

**Fig. 8 | 16 S analysis of the gut microbiota and quantification of microbiota-derived short-chain fatty acids (SCFAs) in chronic colitis mice. A** Colonic microbiota composition of mice at the genus level in different treatment groups ($n$ = 6 mice). **B** PCoA of beta diversity among each group (based on Bray–Curtis metric distances) ($n$ = 6 mice). **C** Alpha diversity boxplot of the colonic microbiota in different treatment groups ($n$ = 6 mice, $p$ = 5.39938 × 10⁻⁶, 2.27489 × 10⁻², 7.59621 × 10⁻⁵, 7.76256 × 10⁻³ and 5.26150 × 10⁻³). Box plots show data as "Min to Max, show all points". The center line represents the median, box limits represents the 25th percentile and the 75th percentile, whiskers represents the minimum to maximum values that are not outliers. **D** Functional prediction of the first 20 KEGG pathways in the colonic microbiota of mice in different treatment groups. **E** Fecal SCFA levels of mice in each group ($n$ = 8 mice). For Acetic acid, $p$ = 7.32000 × 10⁻¹³,

4.16400 × 10⁻¹² and 1.00000 × 10⁻¹⁵; For Propionic acid, $p$ = 5.32416 × 10⁻⁶, 8.24280 × 10⁻¹¹ and 3.61995 × 10⁻³; For Isobutyric acid, $p$ = 1.00000 × 10⁻¹⁵, 1.00000 × 10⁻¹⁵ and 1.00000 × 10⁻¹⁵; For Butyric acid, $p$ = 1.00000 × 10⁻¹⁵, 1.00000 × 10⁻¹⁵ and 1.00000 × 10⁻¹⁵; For Isovaleric acid, $p$ = 1.62200 × 10⁻¹², 7.09258 × 10⁻¹⁰ and 1.72055 × 10⁻¹⁰; For Valeric acid, $p$ = 2.86180 × 10⁻¹⁰, 1.00000 × 10⁻¹⁵ and 1.00000 × 10⁻¹⁵; For Caproic acid, $p$ = 8.04884 × 10⁻¹, 3.25554 × 10⁻² and 1.74329 × 10⁻³. Data are presented as the mean ± SD. $P$ values derived from Brown-Forsythe and Welch ANOVA tests followed by Two-stage linear step-up procedure of Benjamini, Krieger and Yekutieli test (**C**), or One-way ANOVA analysis followed by Tukey's multiple comparisons test (**E**). Source data are provided as a Source Data file. NPs nanoparticles. ACE Abundance-based Coverage Estimator.

could effectively alleviate symptoms of inflammation in melanoma[20], atherosclerosis[39], rheumatoid arthritis[21], and IBD[24]. Based on the favorable biocompatibility, sensitive ROS-responsiveness and good safety of 4-phenylboronic acid pinacol ester-conjugated cyclodextrin biomaterials (Oxi-αCD)[20], we developed a PD-L1-Fc-loaded nano-therapeutic for site-specific delivery of PD-L1-Fc to the inflamed colon in IBD.

After intraperitoneal injection, we found significant localization and long-term accumulation of PD-L1-Fc/Oxi-αCD nanoparticles in the inflamed colon in mice with DSS-induced colitis and less accumulation in other organs. Moreover, a cellular affinity assay revealed that PD-L1-Fc/Oxi-αCD nanoparticles can obviously bind CD4+ T lymphocytes, macrophages, neutrophils and DCs in the colonic lamina propria of colitis mice. PD-L1-Fc/Oxi-αCD nanoparticles significantly improved colitis symptoms, as manifested by decreased weight loss, reduced DAI scores, and a well-maintained colon length as well as notably reduced levels of proinflammatory cytokines and improved colonic morphology and microstructure. Compared to free PD-L1-Fc, PD-L1-Fc/Oxi-αCD nanoparticles had a better effect on acute or chronic colitis. The reason may be that PD-L1-Fc//Oxi-αCD nanoparticles can accumulate in inflammatory tissues and show special affinity toward inflammatory cells.

In IBD patients or mice with DSS-induced colitis, mucosal macrophages can generally produce proinflammatory factors, damage intestinal epithelial cells and weaken the intestinal barrier, leading to many neutrophils being recruited to sites of intestinal lesions, which aggravates IBD progression[40–42]. DCs play a central role in regulating immunity, and PD-L1-Fc may act directly on colonic macrophages and DCs by interacting with upregulated PD-1 to modulate cytokine production[17]. In our study, both PD-L1-Fc and the associated nano-therapeutics effectively reduced the frequencies of macrophages, neutrophils and DCs, as well as the cytokine production of these cell types in mice with DSS-induced colitis. Notably, PD-L1-Fc/Oxi-αCD nanoparticles had the most positive effect on these cells because the nanoparticles targeted the delivery of PD-L1-Fc to these cells.

In addition to innate immune cells, T lymphocytes play a pivotal role in the immune response underlying IBD pathogenesis. Th1 and Th17 cell populations are imbalanced and compete with each other during the development of IBD[43,44]. During the development of DSS-induced colitis, PD-L1-Fc effectively promotes the development of colonic Th1 cells through preferential suppression of Th17 cells[17], which is consistent with the flow cytometry results in our study, and the effect of PD-L1-Fc/Oxi-αCD nanoparticles on Th1 cells was more significant than that of the other treatments. The PD-1/PD-L1 pathway can modulate Treg cell activation and function[45]. In preeclampsia, the PD-1/PD-L1 pathway promotes the Treg/Th17 imbalance[46]. Anti-PD-1 treatment reduces Treg differentiation and infiltration in primary liver cancer[47]. The loss of immune homeostasis secondary to defects in the Treg pool is recognized as a driver of intestinal inflammation[26]. Tfh cells also play a vital role in autoimmune diseases[48]; however, their role in the pathogenesis of IBD is controversial. Cao et al. demonstrated that transfer of Tfh cells could alleviate DSS-induced colitis in mice with ATF3 deficiency in CD4+ T cells[25], whereas the results of Zhang et al. showed that IRF8-regulated Tfh cells could directly cause colon inflammation in an in vivo T-cell transfer animal model of colitis[49]. In our study, the decreased frequencies of Treg and Tfh cells in DSS-induced mice were rescued by either PD-L1-Fc treatment or PD-L1-Fc/Oxi-αCD nanoparticles treatment, and PD-L1-Fc/Oxi-αCD nanoparticles caused more pronounced increases in the frequencies. The above results suggested that CD4+ T lymphocytes might be a target of PD-L1-Fc therapy in colitis. PD-L1-Fc/Oxi-αCD nanoparticles had a better capacity to target these cells and better therapeutic effects on colitis than PD-L1-Fc, which was most likely due to their specific targeting to inflamed areas that resulted in the accumulation of PD-L1-Fc in colon lesions rather than in other metabolic organs.

IBD is associated with an imbalance between symbiotic and potentially pathogenic microorganisms[50,51]. Our study revealed a high abundance of potentially infectious *Escherichia–Shigella* and a low abundance of *Prevotellaceae* in DSS-induced colitis mice, which could partially contribute to the development of colitis. PD-L1-Fc/Oxi-αCD nanoparticles treatment shifted the microbial community status from a dysregulated state to a normal state. SCFAs are associated with the maintenance of mucosal immunity and may reduce the risk of IBD[52–54]. Sun et al. reported that microbiota-derived SCFAs promote IL-10 production by Th1 cells to maintain intestinal homeostasis[55]. Our data showed that treatment with PD-L1-Fc/Oxi-αCD nanoparticles increased the content of SCFAs in the feces of mice with colitis, which was accompanied by an increase in the proportion of some T lymphocytes in the colonic lamina propria, as well as an increase in IL-10 concentrations.

Compared with PD-L1-Fc therapy, PD-L1-Fc/Oxi-αCD nanoparticles treatment did not cause damage to major organs and had little effect on the body weight, organ indexes and routine blood parameters of mice, as well as the susceptibility to *Pseudomonas aeruginosa* infection in the lungs, implying that PD-L1-Fc-loaded Oxi-αCD nanoparticles had fewer side effects and a better safety profile than PD-L1-Fc.

## Methods

All research complies with the relevant ethical regulations. Study protocols were approved by Laboratory Animal Welfare and Ethics Committee of the Army Medical University (Chongqing, China) for all animal experiments.

### Materials

α-Cyclodextrin (α-CD) was supplied by Tokyo Chemical Industry Co., Ltd. (Tokyo, Japan). 1-(3-Dimethylaminopropyl)−3-ethylcarbodiimide hydrochloride (EDC.HCl), 4-(hydroxymethyl) phenylboronic acid pinacol ester (HPAP), 4-dimethylaminopyridine (DMAP), 1,1′-carbonyldiimidazole (CDI), Pluronic F127 (a polyethylene oxide-polypropylene oxide-polyethylene oxide triblock copolymer, or PEO-PPO-PEO) and poly (lactide-co-glycolide) (PLGA, 50:50) with an intrinsic viscosity of 0.50−0.65 were purchased from Sigma–Aldrich Co., LLC (Saint Louis, MO, USA). Lecithin (refined) was supplied by Alfa Aesar (Ward Hill, MA, USA). 1,2-Distearoylsn-glycero-3-phosphoethanolamine-*N*-[succinimidyl (polyethylene glycol)2000] (DSPE-PEG2000-NHS) and hydrophilic sulfo-cyanine5 maleimide were supplied by Xi'an Ruixi Biological Technology Co., Ltd. (Xi'an, China). Then, 50-kDa and 100-kDa Amicon® Ultra ultrafiltration centrifuge columns were supplied by Merck Chemicals Co., Ltd. (Darmstadt, Germany). RPMI 1640 culture medium, fetal bovine serum and Transcription Factor Staining Buffer were purchased from Thermo Fisher Scientific Inc. (Waltham, MA, USA). Dextran sulfate sodium (DSS, 35000 Da) was purchased from MP Biomedicals Co., Ltd. (Solon, Ohio, USA). Cyclophosphamide monohydrate was purchased from Aladdin Biochemical Technology Co., Ltd. (Shanghai, China). The Bicinchoninic Acid (BCA) protein quantification/concentration determination kit, Cell Counting Kit-8 and Coomassie Blue Fast Staining Solution were purchased from Beyotime Biotechnology Co., Ltd. (Hangzhou, China). PD-L1-Fc protein was purchased from NovoProtein Scientific Inc. (Suzhou, China). Precoated ELISA kits for IFN-γ, TNF-α, IL-17A, IL-1β, IL-6 and IL-10 were purchased from Dakewe Bioengineering Co., Ltd. (Shenzhen, China). Collagenase IV was purchased from Worthington Biochemical Co. (Freehold, New Jersey, USA), and Percoll was purchased from GE Healthcare Co., Ltd. (Chalfont St. Giles, Buckinghamshire, UK). DL-Dithiothreitol (DTT), ethylene diamine tetraacetonitrile acid (EDTA), DNase I and the mouse spleen lymphocyte cell isolation kit were purchased from Solarbio Science & Technology Co., Ltd. (Beijing, China). Transcription Factor Staining Buffer was purchased from Invitrogen (Carlsbad, CA, USA).

## Animals, cells and bacteria

Six-week-old female C57BL/6J mice weighing 18–20 g (stock number: N000013) were purchased from Gempharmatech Biotechnology Co., Ltd. (Nanjing, China). The mice were bred and housed under specific-pathogen-free (SPF) conditions and maintained in microisolator cages on individually ventilated cage-racks filled with aspen chip bedding at a room temperature of 20–22 °C in a humidity-controlled (45–65 °C) environment under a 12-h light/dark cycle, with ad libitum access to autoclaved rodent chow and autoclaved water. All diets of mice were provided by Laboratory Animal Science of Army Medical University. Study protocols were approved by the Laboratory Animal Welfare and Ethics Committee of the Army Medical University (approval number: AMUWEC20212167). All mice were acclimatized for one week before further experiments. In the study, the female C57BL/6J mice were used to all experiments. The experimental mice and control mice were bred separately in different cages. All mice were euthanized by cervical dislocation at the end of each animal experiment.

Mouse spleen lymphocyte cells and colonic lamina propria cells were isolated from the spleens and colons of 6-week-old female C57BL/6J mice, respectively. All cells were cultured in RPMI 1640 cell medium with 10% FBS and were incubated at 37 °C under a humidified atmosphere containing 5% $CO_2$. *Pseudomonas aeruginosa* used for the establishment of pneumonia models was purchased from BeNa Culture Collection Co., Ltd. (Xinyang, China).

## Fabrication of nanoparticles

ROS-responsive α-CD (Oxi-αCD) was synthesized according to our previously reported methods[21,23]. Specifically, HPAP (2.77 g, 11.8 mmol) was reacted with CDI (3.83 g, 23.6 mmol) in dry dichloromethane (DCM, 20 mL) to obtain CDI-activated HPAP (3.55 g). ROS-responsive α-CD (Oxi-αCD) was then synthesized by conjugating HPAP onto α-CD. With DMAP (1.0 g, 8.1 mmol) as a catalyst, CDI-activated HPAP (2.0 g, 6.1 mmol) was reacted with α-CD (330 mg, 0.338 mmol) in DMSO (20 mL) and magnetically stirred at room temperature overnight to obtain Oxi-αCD (0.735 g). The obtained Oxi-αCD was characterized by $^1H$ NMR spectroscopy in our previous studies[2,3], which was recorded on a Bruker spectrometer (500 MHz) using deuterated methanol as a solvent.

Nanoparticles loaded with PD-L1-Fc proteins were prepared using a nanoprecipitation self-assembly method[21]. To prepare the PD-L1-Fc/Oxi-αCD nanoparticles, 5 mg of lecithin and 6 mg of DSPE-PEG2000-NHS were dispersed in 0.4 mL of anhydrous ethanol and sonicated for 120 s. Next, 10 mL of ultrapure water was added to the dispersion and sonicated for another 120 s. The aqueous dispersion was slowly stirred in a 65 °C oil bath for 30 min. Then, 50 mg of Oxi-αCD was dissolved in 2.0 mL of methanol and added dropwise to the preheated lipid dispersed solution at a rate of 1 ml/min. The above mixture was rapidly stirred at 800 rpm for 3 min and allowed to self-assemble for 2 h with stirring at 150 rpm. The Blank/Oxi-αCD nanoparticles were obtained by centrifugation at $15,000 \times g$ for 10 min and then washed twice with 10 mL of 5% F127. The solidified nanoparticles were resuspended in 0.2 mL of ultrapure water. Finally, 50 μL of Blank/Oxi-αCD nanoparticles and 250 μg of PD-L1-Fc protein were reacted in an aqueous solution of 300 μL for 48 h (4 °C, pH = 8.5) to prepare PD-L1-Fc/Oxi-αCD nanoparticles. After 48 h, the reaction mixture was added to a 100-kDa ultrafiltration spin column, centrifuged at $4000 \times g$ for 30 min at 4 °C, and then reverse centrifuged at $1000 \times g$ for 5 min to remove the PD-L1-Fc protein that was not loaded on the Oxi-αCD nanoparticles.

The steps to prepare the PD-L1-Fc/PLGA nanoparticles were the same as those to prepare the PD-L1-Fc/Oxi-αCD nanoparticles. Briefly, 9 mg of lecithin and 18 mg of DSPE-PEG$_{2000}$-NHS were dispersed in 0.4 mL of anhydrous ethanol, and 50 mg of Oxi-αCD was replaced by 30 mg of PLGA and dissolved in 1.5 mL of acetonitrile.

To synthesize Cy5-labeled Oxi-αCD (Cy5-Oxi-αCD), 5 mg of Cy5-free acid (0.00963 mmol), 7.4 mg of EDC (0.0385 mmol) and 2.0 mg of DMAP (0.0164 mmol) were dissolved in 5.0 mL of DMF, and then 50 mg of α-CD (0.0514 mmol) was added to the above solution and reacted at room temperature for 2 days. After 2 days, the DMF was removed under reduced pressure, and the residue was washed with acetone 3 times to obtain Cy5-αCD conjugates. Then, 50 mg of Cy5-αCD conjugates and 0.3 g of CDI-activated HPAP (0.915 mmol) were reacted to obtain Cy5-labeled Oxi-αCD by using 0.15 g of DMAP (1.216 mmol) as a catalyst. The steps to prepare the Cy5-Oxi-αCD nanoparticles loaded with PD-L1-Fc proteins (defined as PD-L1-Fc/Cy5-Oxi-αCD nanoparticles) were the same as those to prepare the PD-L1-Fc/Oxi-αCD nanoparticles.

To prepare Cy5-labeled PD-L1-Fc/Oxi-αCD nanoparticles (Cy5-PD-L1-Fc/Oxi-αCD nanoparticles), 0.5 mg of Cy5-mal was slowly dropped into an aqueous solution of 1 mg of PD-L1-Fc protein at pH = 7.0 and then reacted for 24 h at 4 °C. After 24 h, the reaction mixture was loaded into a 50-kDa Amicon® Ultra ultrafiltration spin column and centrifuged at $4000 \times g$ for 30 min at 4 °C to remove unincorporated Cy5-mal labels. Cy5-labeled PD-L1-Fc (Cy5-PD-L1-Fc) was then reacted with Blank/Oxi-αCD nanoparticles for 48 h (4 °C, pH = 8.5) and added to a 100-kDa Amicon® Ultra ultrafiltration spin column to remove the Cy5-PD-L1-Fc that was not loaded on the Blank/Oxi-αCD nanoparticles. The steps to prepare the Cy5-PD-L1-Fc/PLGA nanoparticles were the same as those to prepare the Cy5-PD-L1-Fc/Oxi-αCD nanoparticles.

## Characterization of nanoparticles

The size, polymer dispersity index (PDI) and zeta potential of the nanoparticles were measured by DLS (Zetasizer Nano ZSP, Malvern Instrument, UK). The morphology of the nanoparticles was observed by transmission electron microscopy (TECNAI-10 microscope, Philips, Netherlands) and scanning electron microscopy (FIB-SEM microscope, Crossbeam 340, Zeiss).

The PD-L1-Fc removed by ultrafiltration was measured by the BCA assay, and the PD-L1-Fc protein loading on nanoparticles was calculated by the formula: PD-L1-Fc loading ratio (%) = [(250 μg − the amount of removed protein)/250 μg] × 100%.

To measure the loading content of Cy5-PD-L1-Fc in different nanoparticles, the Cy5-PD-L1-Fc-loaded nanoparticles were dispersed in methanol for extraction of Cy5-PD-L1-Fc. Then, the concentration of Cy5-PD-L1-Fc was quantified by fluorescence spectroscopy (F-7000, Hitachi, Japan) with an excitation wavelength of 628 nm and emission wavelength of 692 nm. The concentration of Cy5-PD-L1-Fc was calculated according to a standard curve established with a series of Cy5-PD-L1-Fc aqueous solutions with predetermined concentrations.

## In vivo biodistribution of Cy5-labeled nanoparticles in mice with acute colitis

Six-week-old female C57BL/6J mice were induced to establish acute colitis model by the addition of 2% (w/v) DSS to the drinking water for 9 days. After 9 days of treatment with DSS, the mice were intraperitoneally injected with Cy5-labeled PD-L1-Fc or Cy5-labeled nanoparticles (the Cy5 dose was 40 μg). The experimental mice and control mice were bred separately in different cages. At 0, 2, 4, 8, 16, 24 or 48 h after injection, the mice were euthanized, and the heart, lung, spleen, liver, kidney, stomach, small intestine and colon tissues were collected to analyze the biodistribution of Cy5-labeled PD-L1-Fc or Cy5-labeled nanoparticles using a live animal imaging system (Biolight Biotechnology Co., Ltd. Guangzhou, China) with a 625-nm excitation filter and a 680-nm emission filter.

To further confirm the localization of nanoparticles in the inflamed colon, we prepared frozen sections of colons from healthy mice and colitis mice treated with PD-L1-Fc/Cy5-Oxi-αCD nanoparticles (containing 40 μg of Cy5) for 8 h and assessed the accumulation of nanoparticles in the colon. After 10 μm-thick colon cryosections were stained with DAPI, images were captured by a confocal laser scanning microscope (LSM880, Zeiss, Germany).

## Immunofluorescence staining for PD-L1-Fc

Healthy female mice and DSS-induced female mice were intraperitoneally injected with human IgG Fc, PD-L1-Fc or PD-L1-Fc nanoparticles (containing 40 µg of PD-L1-Fc). The experimental mice and control mice were bred separately in different cages. After 8 h of treatment, the mice were euthanized, and the colons were isolated to prepare frozen sections. After blocking for 30 min with 5% BSA, frozen colonic sections were incubated with FITC-conjugated anti-human IgG Fc antibody (1:1000) overnight and then stained with DAPI for 15 min after washing with PBS. After washing 3 times with PBS and staining with DAPI for 15 min, the samples were visualized with CLSM. Fluorescence images were analyzed by Image J 1.53 (National Institutes of Health, USA).

## In vivo binding of PD-L1-Fc

Healthy female mice and DSS-induced female mice were intraperitoneally injected with human IgG Fc, PD-L1-Fc or PD-L1-Fc nanoparticles (containing 40 µg of PD-L1-Fc). The experimental mice and control mice were bred separately in different cages. After 8 h of treatment, the mice were euthanized and the colons were removed in cold PBS solution. After removing adipose and mesenteric tissues, the colon was dissected longitudinally along one side of the mesentery, rinsed, and cut into segments of approximately 0.5–1.0 cm laterally. The colonic segments were placed in a washing solution consisting of 1 mM DTT, 5 mM EDTA and 10 mM HEPES, shaken at 250 rpm for 30 min at 37 °C, and filtered through a 100-µm cell strainer. A digestion solution containing 150 µg/mL DNase I and 200 µ/mL collagenase IV was added to the filtered cell suspension and shaken at 250 rpm for 60 min at 37 °C. Then, the cells were filtered through a 100-µm cell strainer and centrifuged at 400 × g for 5 min at room temperature. The cell precipitate was resuspended in 1 mL of 40% Percoll solution, and the lymphocytes were isolated via a Percoll density gradient. The isolated lymphocytes from the colonic lamina propria were then resuspended in PBS and stained with Super Bright 600-conjugated anti-mouse CD3e antibody (1:100), eFluor 450-conjugated anti-mouse CD45 antibody (1:200), PE-Cyanine7-conjugated anti-mouse CD45R antibody (1:200), APC-conjugated anti-mouse CD11b antibody (1:200), FITC-conjugated anti-mouse F4/80 antibody (1:100), FITC-conjugated anti-mouse Ly-6G/Ly-6C antibody (1:100), APC-conjugated anti-mouse CD11c antibody (1:200), FITC-conjugated anti-mouse MHC Class II antibody (1:200) and FITC-conjugated anti-mouse CD4 antibody (1:200) for identifying macrophages (CD11b$^+$F4/80$^+$), neutrophils (CD11b$^+$Ly-6G/Ly-6C$^+$), DCs (CD11c$^+$MHC Class II$^+$) and CD4$^+$ T cells. In addition, cells were stained with PE-conjugated anti-human IgG Fc antibody (1:100) for detecting the binding of PD-L1-Fc with immune cells. All antibodies used in flow cytometry were incubated with cells at 4 °C for 30 min. Flow cytometric analysis was performed on a BD LSRFortessa$^{TM}$ flow cytometer (BD, USA). The data were analyzed using FlowJo v10 software (Treestar, Ashland, OR, USA).

## Therapeutic efficacy of the nanoparticles in DSS-induced colitis

Six-week-old female C57BL/6J mice were given a 2% (w/v) solution of DSS for 9 days to establish an acute colitis model. Healthy mice in the normal group were not treated, while DSS-induced mice in the colitis, Blank/Oxi-αCD nanoparticles, PD-L1-Fc, PD-L1-Fc/Oxi-αCD nanoparticles, and PD-L1-Fc/PLGA nanoparticles groups were injected intraperitoneally with saline, Blank/Oxi-αCD nanoparticles, PD-L1-Fc, PD-L1-Fc/Oxi-αCD nanoparticles, and PD-L1-Fc/PLGA nanoparticles on the 2nd day of DSS feeding, respectively. The dose was 40 µg per mouse of PD-L1-Fc for all PD-L1-Fc-containing formulations. The experimental mice and control mice were bred separately in different cages. Mice were observed daily, and the body weight and disease activity index (DAI) were evaluated. The DAI is defined as the summation of the weight loss score (0–4), the stool consistency score (0–4) and the fecal occult blood score (0–4)[56]. To establish a chronic colitis model, 6-week-old female C57BL/6J mice were provided a 1.5% (w/v) DSS solution on days 1–5, 11–15, and 21–25 and were intraperitoneally injected with saline, Blank/Oxi-αCD nanoparticles, PD-L1-Fc, PD-L1-Fc/Oxi-αCD nanoparticles or PD-L1-Fc/PLGA nanoparticles after each period of DSS administration. The dose was 40 µg per mouse of PD-L1-Fc for all PD-L1-Fc-containing formations. The therapeutic effect was evaluated as described above.

## Mini-endoscopic imaging

An endoscopic video system for mice was used for direct visualization of DSS-induced colonic mucosal damage. The colonoscopy apparatus included a miniature endoscope (oscilloscope with an outer diameter of 1.9 mm), a xenon light source, a camera (Karl Storz, Germany), and an air supply pump to achieve regulated inflation of the mouse colon. Before imaging, mice were anesthetized by inhalation of isoflurane (RWD Life Science, China). The endoscopic procedure was viewed on a color monitor and digitally recorded on tape. The colonoscopy severity score was blindly scored by applying the Murine Endoscopic Index of Colitis Severity (MEICS) based on the granularity of the mucosal surface (0–3), vascular pattern (0–3), translucency of the colon mucosa (0–3), visible fibrin (0–3) and stool consistency (0–3)[57].

## Histological assessment

A segment (1 cm) of the distal colon was fixed in 4% (v/v) buffered formalin and embedded in paraffin. Tissue sections with a thickness of 7 µm were stained with hematoxylin and eosin (H&E). The histology of the colon was evaluated by optical microscopy, and the pathological severity was blindly scored based on inflammatory cell infiltration (0–4), goblet cell depletion or decreased mucus accumulation (0–4), mucosal thickening (0–4), destruction of architecture (0, or 3–4) and loss of crypts (0, or 3–4)[58].

## Scanning electron microscopy imaging of colonic tissues from colitis mice

Colonic tissue (2 cm × 2 cm) from mice with acute or chronic colitis was gently washed 3 times in PBS and then fixed with 0.25% glutaraldehyde. After dehydration, embedding, sectioning and staining, the samples were observed under a scanning electron microscope (Crossbeam 340, Zeiss).

## Quantification of inflammatory factors in colonic tissue

Colon tissues were washed, cut into pieces, mixed with PBS buffer (pH 7.4) at a ratio of 1:9, homogenized for 3–5 min (homogenization: 10–15 s/time, gap: 10 s), and centrifuged at 12,000 × g for 10 min at 4 °C. The supernatant was collected, and the precipitate was discarded. The concentrations of IFN-γ, TNF-α, IL-17A, IL-1β, IL-6 and IL-10 in colon homogenates were determined according to the instructions of the appropriate ELISA kit.

## Flow cytometry analysis of colonic lamina propria cell subgroups

The lymphocytes from colonic lamina propria were stained with Super Bright 600-conjugated anti-mouse CD3e antibody (1:100), eFluor 450-conjugated anti-mouse CD45 antibody (1:200), PE-Cyanine7-conjugated anti-mouse CD45R antibody (1:200), PE-conjugated anti-mouse TCRβ antibody (1:200), FITC-conjugated anti-mouse CD11b antibody (1:400), PE-conjugated anti-mouse F4/80 antibody (1:100), FITC-conjugated anti-mouse Ly-6G/Ly-6C antibody (1:100), APC-conjugated anti-mouse CD11b antibody (1:200), FITC-conjugated anti-mouse CD11c antibody (1:200), APC-conjugated anti-mouse MHC Class II antibody (1:200), FITC-conjugated anti-mouse CD4 antibody (1:200), APC-conjugated anti-mouse FOXP3 antibody (1:20), APC-conjugated anti-mouse CXCR3 antibody (1:40) and APC-conjugated anti-mouse CXCR5 antibody (1:40) for identifying macrophages

(CD11b⁺F4/80⁺), neutrophils (CD11b⁺Ly-6G/Ly-6C⁺), DCs (CD11c⁺MHC Class II⁺), Treg (CD4⁺FOXP3⁺), Th1 (CD4⁺CXCR3⁺) and Tfh (CD4⁺CXCR5⁺) cells. All antibodies used in flow cytometry were incubated with cells at 4 °C for 30 min. The percentages of different cell subgroups were assessed by flow cytometry using a BD LSRFortessa™ flow cytometer.

## Toxicity evaluation of PD-L1-Fc-loaded nanoparticles

On the 10th day of DSS treatment, healthy female mice and DSS-induced mice subjected to saline, Blank/Oxi-αCD nanoparticles, PD-L1-Fc protein or PD-L1-Fc/Oxi-αCD nanoparticles were euthanized. The major organs, including the heart, liver, spleen, lungs, kidneys, stomach and intestine, were isolated and stained with H&E.

In another cohort study, mice were randomly divided into two groups. The experimental group was intraperitoneally injected with PD-L1-Fc/Oxi-αCD nanoparticles at a dose of 5.0 g/kg. The control group was intraperitoneally injected with saline. The experimental mice and control mice were bred separately in different cages. After two weeks, the mice were euthanized. The major organs, including the colon, heart, liver, spleen, lung, kidney, stomach, and intestine, were isolated. Histopathological sections were prepared and stained with H&E.

## Establishment of CTX-induced immunosuppression and pneumonia models

Female C57BL/6J mice aged 6–8 weeks were randomly divided into a normal group and a CTX model group. The experimental group were intraperitoneally injected with 80 mg/kg CTX on days 1, 3 and 5. The control group received an intraperitoneal injection of saline equivalent to the dose of CTX on days 1, 3, and 5. The experimental mice and control mice were bred separately in different cages. On day 4 of model establishment, the mice in each group were injected with 40 μL of saline, 40 μg of PD-L1-Fc, or PD-L1-Fc/Oxi-αCD nanoparticles loaded with 40 μg of PD-L1-Fc protein. The daily changes in the body weight of the mice were observed during the model establishment period. On the 7th day, 10 μL of venous blood was collected from the tail vein for routine blood determination. After the mice were euthanized, the thymus and spleen were isolated, and the organ indexes (organ weight/body weight×100%) were calculated.

*Pseudomonas aeruginosa* at 10⁷ CFU was injected into the nasal cavity of mice treated with normal saline, PD-L1-Fc, or PD-L1-Fc/Oxi-αCD nanoparticles after anesthetization by intraperitoneal injection of 1% pentobarbital (40 mg/kg). The mice were euthanized at 24 h after treatment with *P. aeruginosa*, and lung tissues were isolated for paraffin embedding, sectioning, and H&E staining. The lung injury score, referred to as the Smith score, was calculated as the sum of all individual injury scores[59], comprising the assessment of four parameters: pulmonary edema (0–4), alveolar and interstitial inflammation (0–4), alveolar and interstitial hemorrhage (0–4), pulmonary atelectasis, and hyaline membrane formation (0–4).

## SCFA analysis

Fecal samples (0.1 g) were placed in a 2 mL centrifuge tube, and 50 μL of 15% phosphoric acid was added, followed by 100 μL of 125 μg/ml internal standard (isohexanoic acid) solution and 400 μL of diethyl ether homogenate, and incubated for 1 min. The feces were centrifuged at 12,000 × g and 4 °C for 10 min, and the supernatant was taken for quantitative analysis of acetic acid, propionic acid, butyric acid, isobutyric acid, valeric acid, isovaleric acid, and hexanoic acid via a Thermo Trace 1300 gas chromatograph equipped with a Thermo ISQ 7000 mass spectrometric detector (Waltham, MA, USA).

An Agilent HP-Innowax capillary column (30 m × 0.25 mm × 0.25 μm) was used for split injection. The injection volume was 1 μL, and the split ratio was 10:1. The inlet temperature was 250 °C, the ion source temperature was 300 °C, and the transmission line temperature

was 250 °C. The starting temperature of the heating program was 90 °C; then, the temperature was raised to 120 °C at 10 °C/min. Then, the temperature was raised to 150 °C at 5 °C/min. Finally, the temperature was raised to 250 °C for 2 min at 25 °C/min. The carrier gas was helium with a flow rate of 1.0 mL/min. Eight mice from each group were used for SCFA analysis.

## Analysis of bacterial community composition

Stool samples from each group were subjected to 16S rRNA gene amplicon sequencing using Illumina HiSeq 2000 and an established pipeline. 16S rRNA sequence data were analyzed via QIIME2 (2021.11). Thirty amplicon sequence variants (ASVs) and Taxonomic classifications were performed with the DADA2 R package (version 1.30.0). Differences in bacterial community composition were analyzed with DESeq2 (Version 1.42.0)..Six mice from each group were used for 16S rRNA gene amplicon analysis.

## Statistics and reproducibility

Female mice were used for all the animal assays. All experiments contained at least three independent replicates. All data are presented as the mean ± standard deviation (SD). The exact sample size and the statistical tests are described in figure legends, exact $p$ values are provided in the figure legends and Source Data. No statistical method was used to predetermine the sample size. All statistical analysis were performed using one-way ANOVA, two-way ANOVA, Brown-Forsythe and Welch ANOVA tests or two-tailed Student's $t$ test by GraphPad Prism software 8.0 (San Diego, CA, USA).

## Reporting summary

Further information on research design is available in the Nature Portfolio Reporting Summary linked to this article.

## Data availability

The raw 16S rRNA gene sequences data generated in this study have been deposited in the Genome Sequence Archive (GSA) database[60], under accession code CRA014490. The authors declare that all data provided in this study are available within the Article, Supplementary Information or Source data file. Source data are provided with this paper.

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

## Acknowledgements
The authors acknowledge the support provided by the Key Support Object of Army Medical University (410301060191 to D.Z.).

## Author contributions
X.D.T., Y.Y.S. and H.Y. performed the major experiments and prepared the figures and tables. Y.L.S. and S.L. participated in the animal experiments and cell experiments. Y.S.Q. participated in the analysis of the experimental data. J.Y.S., K.C. and Y.L. participated in data curation and provided valuable suggestions for revising the manuscript. L.C. and D.L.Z. designed the project and modified the manuscript. All the authors approved the final manuscript.

## Competing interests
The authors declare no competing interests.
