## [Peer Review File · Nature Communications]

Targeted delivery of Fc-fused PD-L1 for effective management of acute and chronic colitisREVIEWER COMMENTS

Reviewer #1 (Remarks to the Author):

In this manuscript, the authors describe the production of nanoparticles containing PD-L1-Fc that are specifically activated by reactive oxygen species and characterize their impact on acute and chronic DSS colitis.

While this is a conceptionally interesting targeted approach, numerous questions remain open.

- Most importantly, the focus of the manuscript is on the description of the colitis models, but the actually important point of the nanoparticle construction is only superficially addressed. It does not appear in the main methods nor in the main figures. Supplementary figure 1 shows some representative data including inadequately labeled microscopic pictures and Western Blot, but this is not at all sufficient to prove that indeed PD-L1 has been loaded onto these nanoparticles. Moreover, it is unclear whether the production process preserves the specificity for reactive oxygen species. Thus, it is completely unclear what kind of compound the experimental colitis experiments actually investigate and to which mechanisms the effects relate.
- Fig. 1 is rather a graphical abstract than a figure. There are no data on IBD (which is a human disease) in the manuscript, thus the term should be avoided.
- Fig. 2 G, H, I: quantification needed; the endoscopy images are not very convincing. They are either too bright or too dark, I don't see any fibrin or stool, but localized bleeding that could also result from injury by the endoscope
- Fig. 3 F, G, H: the same applies
- Fig. 2J & 3I: Although there are clear clinical differences between free PD-L1 and NP PD-L1, there are rather small differences in cytokine expression - how can that be explained?
- Fig. 4 & 5: Staining and gating are not at all convincing, in many cases the gates do not contain clear populations and the gating strategy is unclear.
- "Safety data": The authors claim superior safety, but in mice with colitis the only systemic difference is a reduction of the white pulp.
- Fig. 6 A, B, C: quantification is needed
- Fig. 7: Details on mating and housing of the mice are required to allow appropriate interpretation of the data
- Suppl. Fig. 3: Quantification is needed
- I.130 - not supported by any data in the context of colitis

Reviewer #2 (Remarks to the Author):

In this manuscript, the authors developed a ROS-responsive nanoparticles to deliver a Fc-fused PD-L1 protein to the inflammatory site at colon for colitis treatment. The authors comprehensively evaluated the efficacy and biosafety of this nanosystem in DSS-induced mouse models. It is a valuable attempt to use nano-delivery system to expand the therapeutic windows of biologics drugs for inflammatory diseases where fine tuning of beneficial and adverse effect is an art.

Although the biofunction and preliminary toxicity results are encouraging, how the ROS-responsive system work and what is the connection with the therapeutic advantages are not well characterized. Several key studies are needed to support the claims. Before those fundamental questions are resolved, the current manuscript is not suitable for publish.

1. It is not clear what is the mechanism for nanoparticle that could selectively accumulate in the inflammatory site with less exposure to rest of body, which is the key advantage of their system over free PD-L1-Fc. A more quantitative biodistribution measurement would help to support this claim.

2. A hypothesis is that the ROS environment may trigger the release of protein drug from the NP, while in no ROS environment, the PD-L1-Fc will remain attached on the NP and stay non-

functional. However, the fluorescent-labeling results could not tell whether the PD-L1-Fc is on the particle or being released free. Also, there is no data provided that if PD-L1-Fc is functional on the NP or muted. Without that group, the necessity of ROS responsive ability would not be reflected. A group of nanoparticles with similar structure of PD-L1-Fc/Oxi- α CD but without the ROS responsive ability should be added.

3. The author claimed that in ROS high environment, the nanoparticle would degrade due to the degradation of the 4-phenylboronic acid pinacol ester. But the author did not explain the mechanism of the nanoparticle assembly and why the degradation of 4-phenylboronic acid pinacol ester could lead to the whole nanoparticle degradation. More extensive in vitro physical-chemical characterization is needed to demonstrate this ROS trigger response reaction.

4. Stability and bioactivity of the PD-L1-Fc after chemical conjugation to the NP and after released free from NP need to be confirmed. Especially there is a concern that the non-specific NHS ester could react with -NH₂ residuals on the binding site of the protein.

5. It is not clear whether the ROS produced by inflammatory cell is existed inside or outside of the cell. If the ROS cleavage is happened intracellularly (as illustrated in Figure 1) after the nanoparticles are internalized, how could the PD-L1-Fc cargo be excreted outside of cell? Any data to support? In addition, nanoparticles in vivo mostly are cleared non-specifically by mononuclear phagocytic system, e.g Kupffer cells in liver and monocytes in spleen, the PD-L1-Fc bind to the NP will be cleared as well, which will not be functional. Fluorescent labeling study could not tell if the protein drugs are free in intercellular space or being trapped inside of endosomes.

Some other minor comments:

- In the cellular uptake experiment, as shown in Figure S2, the number of different immune cells is too poor to make the data unreliable, especially the dendritic cells.
- In the animal experiments, the number of mice should be increased to at least five.
- The figures are not uniform in size, and diagrams of supplementary information are incorrectly placed in the manuscript.
- The ordinate units of Figure 2J and 3I are wrong, they should be "pg/mL".
- The legends of Figure 4A-F and 5A-F are chaotic.

Thank you for your letter and the reviewers' comments concerning our manuscript titled "Targeted delivery of Fc-fused PD-L1 for effective management of acute and chronic colitis" (Manuscript Number: NCOMMS-23-02939). We have studied the comments carefully and have made corrections that we hope will be met with approval. The revised portions are marked in red in the revised manuscript. The main corrections in the paper and the responses to the reviewers' comments are as follows:

Reviewer #1 (Remarks to the Author):

In this manuscript, the authors describe the production of nanoparticles containing PD-L1-Fc that are specifically activated by reactive oxygen species and characterize their impact on acute and chronic DSS colitis.

While this is a conceptually interesting targeted approach, numerous questions remain open.

- Most importantly, the focus of the manuscript is on the description of the colitis models, but the actually important point of the nanoparticle construction is only superficially addressed. It does not appear in the main methods nor in the main figures. Supplementary figure 1 shows some representative data including inadequately labeled microscopic pictures and Western Blot, but this is not at all sufficient to prove that indeed PD-L1 has been loaded onto these nanoparticles. Moreover, it is unclear whether the production process preserves the specificity for reactive oxygen species. Thus, it is completely unclear what kind of compound the experimental colitis experiments actually investigate and to which mechanisms the effects relate.

Response: Fig. 1 in the manuscript illustrates the structure of the nanoparticles. The preparation methods of the nanoparticles are described in the "Methods" section of this manuscript. To confirm that PD-L1-Fc was successfully loaded on the surface of the nanoparticles, we separated free proteins from nanoparticles by using ultrafiltration devices (molecular weight cutoff: 100 kDa) to purify the PD-L1-Fc-loaded nanoparticles. In addition, the PD-L1-Fc protein loaded on the nanoparticles was identified by SDS-PAGE protein electrophoresis and western blot assays. As shown in Fig. 2C-2D and Supplementary Fig. 2C-2D, the molecular weight of PD-L1-Fc on the

PD-L1-Fc/Oxi- α CD NPs or PD-L1-Fc/PLGA NPs was consistent with that of free PD-L1-Fc. Furthermore, the particle size and zeta potential of PD-L1-Fc-loaded nanoparticles were significantly higher than those of the nanoparticles without PD-L1-Fc loading (Supplementary Table 1). All the experimental results demonstrated that PD-L1-Fc was successfully loaded on the surface of the nanoparticles.

We also measured the light transmittance at 500 nm of the blank Oxi- α CD NPs and PD-L1-Fc/Oxi- α CD NPs with or without 1.0 mM H₂O₂ incubation and plotted the hydrolysis curve of these nanoparticles, which confirmed that these nanoparticles can be degraded in a reactive oxygen species microenvironment (Supplementary Fig. 1A-1B).

- Fig. 1 is rather a graphical abstract than a figure. There are no data on IBD (which is a human disease) in the manuscript, thus the term should be avoided.

Response: Thank you for your constructive feedback. Indeed, our research was conducted on mouse models, and we have made the necessary revisions to our manuscript based on your suggestions. In addition, Fig. 1 is referenced in the manuscript as an illustration of the research work presented within the manuscript and can also be described as a graphical abstract.

- Fig. 2 G, H, I: quantification needed; the endoscopy images are not very convincing. They are either too bright or too dark, I don't see any fibrin or stool, but localized bleeding that could also result from injury by the endoscope

Response: Thank you for your valuable comments. We repeated the mouse colitis experiment and obtained new endoscopic images. Due to the limited resolution of the miniature endoscope lens, the images may not meet expectations, but the quality of the images is better than that of the previous images in the manuscript. Because mouse feces can blur the endoscope lens, we performed an enema with saline solution on the mice before conducting the colonoscopy, which ensured that there were no feces in the mouse colons. Due to the limited resolution of the endoscope, the fibrotic lesions in the mouse colon were not clearly visible. The tip of the miniature endoscope we used is not too hard, and there is little possibility of direct damage to the colon mucosa of mice. Moreover, only inflammatory intestinal mucosa will have a brittle texture and be easily subject to scratching. We have replaced the images accordingly.

Murine endoscopy and grading of inflammatory changes were quantified by the murine endoscopic index of colitis severity (MEICS) (please see the revised manuscript). Histological scoring of inflammation was performed on H&E-stained sections (n=6 per group) in a blinded fashion by two independent observers evaluating one whole tissue section per sample. The sum of scores for inflammatory cell infiltration (score, 0–4), goblet cell depletion or decreased mucus accumulation (score, 0–4), mucosal thickening (score, 0–4), destruction of architecture (score, 0 or 3–4) and loss of crypts (score, 0 or 3–4) was calculated (maximum 20). All data are presented as the mean \pm SD (please see the revised manuscript).

We were unable to quantify Fig. 2I (Fig. 3I in the revised version) because we could not find any relevant scoring criteria, and the field was magnified so much that quantification was inaccurate.

- Fig. 3 F, G, H: the same applies

Response: Thank you for your kind reminder. We have revised these figures as above.

- Fig. 2J & 3I: Although there are clear clinical differences between free PD-L1 and NP PD-L1, there are rather small differences in cytokine expression - how can that be explained?

Response: Inflammatory cytokines, inflammatory cells, mucosal repair and other factors are associated with the initiation and progression of colitis. In this study, we assessed not only cytokine secretion in the colonic tissues of mice but also the mini-endoscopic score, histological score, inflammatory cell frequency, etc. The changes in the above indicators between the free PD-L1 and NP PD-L1 groups were consistent with the clinical differences. When we measured the cytokine concentrations, the tissue homogenates were diluted with PBS buffer at a ratio of 1:9. Therefore, after dilution, the numerical differences between the two groups were not as significant as expected. However, there was a statistically significant difference between the two groups.

- Fig. 4 & 5: Staining and gating are not at all convincing, in many cases the gates do not contain clear populations and the gating strategy is unclear.

Response: We only used two markers for flow cytometry analysis in the initial manuscript, making it difficult to completely separate the cell populations. Therefore, to determine cell subsets in isolated colonic lamina propria cells from DSS-induced mice more clearly, we utilized multiple

markers for labeling cell populations. The staining and gating strategy for flow cytometry and the experimental results have been marked in red in the revised manuscript. Please review those modifications.

The gating strategies used to identify macrophages, neutrophils, dendritic cells, Tregs, Th1 cells, and Tfh cells were as follows:

Figure 1. Gating strategy used to identify macrophages, neutrophils, and dendritic cells. First, samples were roughly gated to remove debris based on FSC and SSC. Then, doublets were excluded by FSC-A/SSC-A. Next, samples were gated on the CD45-positive cells for pan-leukocytes, followed by gating on CD3e- and CD45R- negative cells to exclude T cells and B cells. Last, among CD3e- and CD45R- negative cells, F4/80⁺ CD11b⁺ cells (macrophages), CD11b⁺ Ly-6G/Ly-6C⁺ cells (neutrophils) and CD11c⁺ MHC Class II⁺ cells (dendritic cells) can be determined.

Figure 2. Gating strategy used to identify Treg, Th1, and Tfh cells. First, the samples were roughly gated on lymphocytes based on FSC and SSC. Then, doublets were excluded by FSC-A/FSC-H. Next, we gated on the CD45-positive cells for pan-leukocytes and then gated on CD3e and TCR β to differentiate $\alpha\beta$ T cells from non-T cells and non- $\alpha\beta$ T cells. Finally, among $\alpha\beta$ T cells, CD4⁺ FOXP3⁺ T cells (Treg), CD4⁺ CXCR3⁺ T cells (Th1) and CD4⁺ CXCR5⁺ T cells (Tfh) can be determined.

- "Safety data": The authors claim superior safety, but in mice with colitis the only systemic difference is a reduction of the white pulp.

Response: In the safety experiments, we found that compared with the saline group with bacterial infection, a destroyed, incomplete alveolar wall structure and more erythrocytes in the alveolar space were observed in the lungs of PD-L1-Fc-treated mice with bacterial infection. Interestingly, the lungs of the PD-L1-Fc/Oxi- α CD NP-treated mice infected with the bacteria were similar to those of the saline group. In addition, intraperitoneal injection of PD-L1-Fc resulted in a significant reduction in the white blood cell population, neutrophils, lymphocytes and monocytes, but intraperitoneal injection of PD-L1-Fc/Oxi- α CD NPs did not cause this reductions (Figure 6). These results all proved that PD-L1-Fc/Oxi- α CD NPs displayed better safety than PD-L1-Fc.

- Fig. 6 A, B, C: quantification is needed

Response: Please refer to the quantitative results in Fig. 7A, Fig. 7B and Fig. 7C.

- Fig. 7: Details on mating and housing of the mice are required to allow appropriate interpretation of the data

Response: We described the animal experiments in detail in the revised manuscript. Six-week-old female C57BL/6J mice (18-20 g) were purchased from Gempharmatech Biotechnology Co., Ltd. (Nanjing, China). The mice were housed under SPF conditions; maintained in microisolator cages on individually ventilated cage racks in cages filled with aspen chip bedding at 25 °C in a humidity-controlled environment under a 12-h light/dark cycle; and had *ad libitum* access to autoclaved rodent chow and autoclaved water.

- Suppl. Fig. 3: Quantification is needed

Response: Please refer to the quantitative results in Fig. 2H, 2I and Supplementary Fig. 4A-4D.

- l.130 - not supported by any data in the context of colitis

Response: We apologize that we do not understand this question.

Reviewer #2 (Remarks to the Author):

In this manuscript, the authors developed a ROS-responsive nanoparticles to deliver a Fc-fused PD-L1 protein to the inflammatory site at colon for colitis treatment. The authors comprehensively evaluated the efficacy and biosafety of this nanosystem in DSS-induced mouse models. It is a valuable attempt to use nano-delivery system to expand the therapeutic windows of biologics drugs for inflammatory diseases where fine tuning of beneficial and adverse effect is an art.

Although the biofunction and preliminary toxicity results are encouraging, how the ROS-responsive system work and what is the connection with the therapeutic advantages are not well characterized. Several key studies are needed to support the claims. Before those fundamental questions are resolved, the current manuscript is not suitable for publish.

1. It is not clear what is the mechanism for nanoparticle that could selectively accumulate in the inflammatory site with less exposure to rest of body, which is the key advantage of their system

over free PD-L1-Fc. A more quantitative biodistribution measurement would help to support this claim.

Response: The mechanism by which nanoparticles selectively accumulate at inflammatory sites mainly includes passive targeting and active targeting.

1) Passive targeting: In the inflammatory environment, the blood vessels around the inflamed site proliferate and dilate, resulting in enhanced vascular permeability^[1-4]. When nanoparticles pass through the blood vessels of the inflammation site, they can leak through the enlarged vascular space and accumulate in the inflammation site^[5-7]. Nanoparticles modified with polyethylene glycol (PEG) can inhibit the adsorption of serum proteins and escape from the reticulo-endothelial system^[8-11], so the circulation time of the nanoparticles in the blood was prolonged compared with that of free PD-L1-Fc. As shown in Fig. 2H, the accumulation of PD-L1-Fc/Oxi- α CD NPs in the colon of colitis mice was greater than that of PD-L1-Fc, which may be explained by the passive targeting mechanism.

2) Active targeting: Active targeted drug delivery is based on ligand affinity for receptors^[12,13]. Immune cells such as activated macrophages and lymphocytes are abundant at the sites of inflammation, and PD-1 is predominantly expressed on the surface of these cells^[14]; this PD-1 can bind to free PD-L1-Fc or PD-L1-Fc/Oxi- α CD NPs to increase the targeting ability of NPs to inflamed sites.

As shown in Fig. 2H, only weak fluorescence intensity could be detected at 24 h after intraperitoneal injection of free Cy5-PD-L1-Fc, while moderate fluorescence intensity could still be detected at 48 h after intraperitoneal injection of Cy5-PD-L1-Fc/Oxi- α CD NPs. As shown in Supplemental Fig. 4A-4D, Cy5-PD-L1-Fc/Oxi- α CD NPs exhibited lower nonspecific fluorescence distributions than Cy5-PD-L1-Fc in the intestine, liver, spleen, and kidneys. These results demonstrated that PD-L1-Fc/Oxi- α CD NPs can deliver PD-L1-Fc to the inflamed site of the colon in a targeted manner.

References

- [1] Danese S. Role of the vascular and lymphatic endothelium in the pathogenesis of inflammatory bowel disease: 'brothers in arms'. *Gut*. 2011;60(7):998-1008.
- [2] Park-Windhol C, D'Amore PA. Disorders of Vascular Permeability. *Annu Rev Pathol*. 2016;11:251-281.

- [3] Tolstanova G, Deng X, French SW, et al. Early endothelial damage and increased colonic vascular permeability in the development of experimental ulcerative colitis in rats and mice. *Lab Invest.* 2012;92(1):9-21.
- [4] Taniguchi T, Inoue A, Okahisa T, et al. Increased angiogenesis and vascular permeability in patient with ulcerative colitis. *Gastrointest Endosc.* 2009;69:ab365.
- [5] Youshia J, Lamprecht A. Size-dependent nanoparticulate drug delivery in inflammatory bowel diseases. *Expert Opin Drug Deliv.* 2016;13(2):281-294.
- [6] Tu Z, Zhong Y, Hu H, et al. Design of therapeutic biomaterials to control inflammation. *Nat Rev Mater.* 2022;7(7):557-574.
- [7] Durymanov M, Kamaletdinova T, Lehmann SE, Reineke J. Exploiting passive nanomedicine accumulation at sites of enhanced vascular permeability for non-cancerous applications. *J Control Release.* 2017;261:10-22.
- [8] Shi L, Zhang J, Zhao M, et al. Effects of polyethylene glycol on the surface of nanoparticles for targeted drug delivery. *Nanoscale.* 2021;13(24):10748-10764.
- [9] Suk JS, Xu Q, Kim N, Hanes J, Ensign LM. PEGylation as a strategy for improving nanoparticle-based drug and gene delivery. *Adv Drug Deliv Rev.* 2016;99(Pt A):28-51.
- [10] Zhou H, Fan Z, Li PY, et al. Dense and Dynamic Polyethylene Glycol Shells Cloak Nanoparticles from Uptake by Liver Endothelial Cells for Long Blood Circulation. *ACS Nano.* 2018;12(10):10130-10141.
- [11] Zhao C, Deng H, Xu J, et al. "Sheddable" PEG-lipid to balance the contradiction of PEGylation between long circulation and poor uptake. *Nanoscale.* 2016;8(20):10832-10842.
- [12] Zhang Y, Wang Y, Li X, Nie D, Liu C, Gan Y. Ligand-modified nanocarriers for oral drug delivery: Challenges, rational design, and applications. *J Control Release.* 2022;352:813-832.
- [13] Sharpe AH, Wherry EJ, Ahmed R, Freeman GJ. The function of programmed cell death 1 and its ligands in regulating autoimmunity and infection. *Nat Immunol.* 2007;8(3):239-245.
- [14] Boussiotis VA. Molecular and Biochemical Aspects of the PD-1 Checkpoint Pathway. *N Engl J Med.* 2016;375(18):1767-1778.

2. A hypothesis is that the ROS environment may trigger the release of protein drug from the NP, while in no ROS environment, the PD-L1-Fc will remain attached on the NP and stay

non-functional. However, the fluorescent-labeling results could not tell whether the PD-L1-Fc is on the particle or being released free. Also, there is no data provided that if PD-L1-Fc is functional on the NP or muted. Without that group, the necessity of ROS responsive ability would not be reflected. A group of nanoparticles with similar structure of PD-L1-Fc/Oxi- α CD but without the ROS responsive ability should be added.

Response: This is a good question. To address this question, we prepared non-ROS-responsive PD-L1-Fc-loaded NPs (PD-L1-Fc/PLGA NPs). *In vitro* cell experiments showed that both hydrolyzed and unhydrolyzed PD-L1-Fc/Oxi- α CD NPs, as well as PD-L1-Fc/PLGA NPs (without ROS-responsive properties), can bind to activated lymphocytes. The bioactivity of PD-L1-Fc from unhydrolyzed PD-L1-Fc/Oxi- α CD NPs and PD-L1-Fc/PLGA NPs was lower than that of hydrolyzed PD-L1-Fc/Oxi- α CD NPs, which had the same bioactivity as free PD-L1-Fc (Fig. 2E and Supplementary Fig. 2H). The bioactivity of PD-L1-Fc from different formulations is consistent with its effect on the inhibition of lymphocyte proliferation (Fig. 2F and Supplementary Fig. 2I). These results suggested that PD-L1-Fc was successfully loaded on the NPs, and only the released PD-L1-Fc retained its bioactivity.

Flow cytometry analysis was performed to measure the interaction between the NPs and colonic lamina propria lymphocytes from colitis mice by using human PE-conjugated Fc antibody. The results confirmed that the PD-L1-Fc released from PD-L1-Fc/Oxi- α CD NPs can more effectively bind to colonic lamina propria lymphocytes, such as CD4⁺ T cells, dendritic cells, macrophages and neutrophils, than PD-L1-Fc/PLGA NPs (Supplementary Fig. 5 and Supplementary Fig. 7A-7D). Treatment of mice with acute and chronic enteritis also showed that PD-L1-Fc/Oxi- α CD NPs had a better therapeutic effect than PD-L1-Fc/PLGA NPs (Supplementary Fig. 8 and Supplementary Fig. 9). The above results suggest that PD-L1-Fc/Oxi- α CD NPs were hydrolyzed in the ROS microenvironment at the inflamed site and released PD-L1-Fc to bind to inflammatory cells, which led to the PD-L1-Fc/Oxi- α CD NP group displaying a better outcome than the PD-L1-Fc/PLGA NP group with respect to colitis treatment.

3. The author claimed that in ROS high environment, the nanoparticle would degrade due to the degradation of the 4-phenylboronic acid pinacol ester. But the author did not explain the mechanism of the nanoparticle assembly and why the degradation of 4-phenylboronic acid pinacol

ester could lead to the whole nanoparticle degradation. More extensive *in vitro* physical-chemical characterization is needed to demonstrate this ROS trigger response reaction.

Response: This is an interesting question. In our study, we engineered ROS-responsive nanoparticles with a core-shell structure by a self-assembly approach. The NPs are formed from three biomaterials: 1) ROS-responsive α -cyclodextrin (Oxi- α CD), which is synthesized by conjugating 4-phenylboronic acid pinacol ester (HPAP) onto hydroxyl groups, was selected for the hydrophobic core; 2) lecithin was selected as a monolayer around the hydrophobic core; and 3) biodegradable polymer — 1,2-distearoylsn-glycero-3-phosphoethanolamine-*N*-[succinimidyl (polyethylene glycol)2000] (DSPE-PEG2000-NHS) — was interspersed in the lecithin monolayer to form a shell around the hydrophobic core, which can conjugate with PD-L1-Fc. As shown in Scheme 1, the ROS-responsive unit can be degraded under H₂O₂ treatment^[1], which may lead to the degradation of the core-shell polymer backbone. Thus, the entire nanoparticles can be degraded upon exposure to ROS.

4-(Hydroxymethyl) phenylboronic acid pinacol ester (HPAP) has been widely used to modify saccharides (such as dextran and cyclodextrin), polyesters and other polymers to prepare ROS-responsive carriers^[2]. Our team has prepared a series of ROS-responsive nanodrugs^[3-8]. Our previous work also synthesized HPAP-modified cyclodextrin and used it as a carrier to deliver docetaxel, rapamycin, moxifloxacin, dexamethasone, cefpodoxime proxetil, and luteolin to diseased sites^[3-8]. ROS-responsive materials can be degraded at different concentrations of H₂O₂ (0.01, 0.05, 0.25, 0.5 and 1.0 mM H₂O₂)^[3]. The release mechanism of drug-loaded NPs was also discussed. These drug-loaded NPs showed disappearance of their original spherical morphology before *in vitro* drug release (Figure 1A). However, the spherical structure of the NPs collapsed, and drug crystals were observed after drug release (Figure 1B)^[3]. In the present work, we aimed to develop a ROS-responsive nanomedicine for the targeted treatment of colitis and investigated these therapeutic mechanisms. Therefore, the release mechanism of NPs was not discussed in detail.

Editorial Note: Figure 1 below reproduced from Zhang, D., Wei, Y., Chen, K., Zhang, X., Xu, X., Shi, Q., Han, S., Chen, X., Gong, H., Li, X. and Zhang, J. (2015), Biocompatible Reactive Oxygen Species (ROS)-Responsive Nanoparticles as Superior Drug Delivery Vehicles. *Adv. Healthcare Mater.*, 4: 69-76. <https://doi.org/10.1002/adhm.201400299>, with permission from John Wiley and Sons. © 2014 WILEY-VCH Verlag GmbH & Co. KGaA, Weinheim.

Scheme 1. Degradation of Oxi- α CD materials in H_2O_2 .

Figure 1. (A) Atomic force microscopy (AFM) image of drug-loaded Oxi- β CD NPs before *in vitro* drug release. (B) AFM image of drug-loaded Oxi- β CD NPs after drug release. (Copied from our previously published work: *Adv. Healthcare Mater.* 2015, 4, 69-76, Supporting Information, Figure S9).

In Supplementary Fig. 1C, Supplementary Fig. 1E and Supplementary Fig. 2G, we measured the size of the PD-L1-Fc/Oxi- α CD NPs and PD-L1-Fc/PLGA NPs in the presence or absence of H_2O_2 . The PD-L1-Fc/Oxi- α CD NPs exhibited a significantly decreased particle size after H_2O_2 treatment, while the PD-L1-Fc/PLGA NPs did not. These results demonstrated that ROS triggered the hydrolysis of PD-L1-Fc/Oxi- α CD NPs.

[1] Fang-Yi Qiu, Cheng-Cheng Song, Mei Zhang, Fu-Sheng Du, Zi-Chen Li. *ACS Macro Lett.* 2015, 4, 1220-1224.

[2] Christos Tapeinos, Abhay Pandit. *Adv. Mater.* 2016, 28, 5553–5585.

[3] Dinglin Zhang, Yanling Wei, Kai Chen, Xiangjun Zhang, Xiaoqiu Xu, Qing Shi, Songling Han, Xin Chen, Hao Gong, Xiaohui Li, Jianxiang Zhang. *Adv. Healthcare Mater.* 2015, 4,

- 69-76.
- [4] Yu Wang, Qian Yuan, Wei Feng, Wendan Pu, Jun Ding, Hongjun Zhang, Xiaoyu Li, Bo Yang, Qing Dai, Lin Cheng, Jinyu Wang, Fengjun Sun, Dinglin Zhang. *J Nanobiotechnol* (2019) 17:103, 1-16.
- [5] Yin Dou, Yue Chen, Xiangjun Zhang, Xiaoqiu Xu, Yidan Chen, Jiawei Guo, Dinglin Zhang, Ruibing Wang, Xiaohui Li, Jianxiang Zhang. *Biomaterials* 143 (2017) 93-108.
- [6] Jun Zheng, Ruimin Hu, Yang Yang, Yu Wang, Qianmei Wang, Senlin Xu, Pu Yao, Zhiyong Liu, Jiangling Zhou, Jing Yang, Ying Bao, Dinglin Zhang, Wenhao Shen, Zhansong Zhou, Antibiotic-loaded reactive oxygen species-responsive nanomedicine for effective management of chronic bacterial prostatitis. *Acta Biomaterialia*. 2022, 43, 471-486.
- [7] Yu Wang, Qianmei Wang, Wei Feng, Qian Yuan, Xiaowei Qi, Sheng Chen, Pu Yao, Qing Dai, Peiyuan Xia, Dinglin Zhang, Fengjun Sun. Folic acid-modified ROS-responsive nanoparticles encapsulating luteolin for targeted breast cancer treatment. *Drug Delivery*, 2021, 28(1):1695-1708.
- [8] Rongrong Ni, Guojing Song, Xiaohong Fu, Ruifeng Song, Lanlan Li, Wendan Pu, Jining Gao, Jun Hu, Qin Liu, Fengtian He, Dinglin Zhang, Gang Huang. Reactive oxygen species-responsive dexamethasone-1 loaded nanoparticles for targeted treatment of rheumatoid arthritis via suppressing the iRhom2/TNF- α /BAFF signaling pathway. *Biomaterials*, 2020, 232: 119730.

4. Stability and bioactivity of the PD-L1-Fc after chemical conjugation to the NP and after released free from NP need to be confirmed. Especially there is a concern that the non-specific NHS ester could react with -NH₂ residuals on the binding site of the protein.

Response: This is a good question. The PD-L1-Fc/Oxi- α CD NPs and PD-L1-Fc/PLGA NPs displayed a constant particle size and PDI after 24 h of incubation with 10% fetal bovine serum (FBS) medium at 37 °C (Supplementary Fig. 3A and 3B), implying that these nanoparticles can maintain a certain stability in serum. Then, the NPs subjected to 0-h or 24-h incubation in 10% FBS medium were used to treat anti-CD3 activated lymphocytes for 8 h. Flow cytometry analysis showed no obvious changes in PD-L1-Fc binding to lymphocytes between the NPs incubated for 0 h and the NPs incubated for 24 h (Supplementary Fig. 3C-3D), further demonstrating that

PD-L1-Fc-loaded NPs maintain a certain stability in serum.

As the reviewer mentioned, PD-L1-Fc was conjugated with DSPE-PEG-NHS through amide bonds. We only delivered PD-L1-Fc to immune cells and did not need to break the amide bond. Consequently, maintaining the bioactivity of PD-L1-Fc conjugated to DSPE-PEG-NHS became very important. Therefore, we detected the bioactivity of PD-L1-Fc conjugated to DSPE-PEG-NHS, which was released from the PD-L1-Fc/Oxi- α CD NPs. As shown in Fig. 2E and Supplementary Fig. 2H, the human PE-conjugated Fc antibody was used to detect the bioactivity of PD-L1-Fc in the PD-L1-Fc/Oxi- α CD NPs and PD-L1-Fc/PLGA NPs. The results showed that both free PD-L1-Fc and PD-L1-Fc in the NPs (PD-L1-Fc conjugated to DSPE-PEG-NHS) can bind to activated lymphocytes. Importantly, PD-L1-Fc released from the PD-L1-Fc/Oxi- α CD NPs (which indeed released the PD-L1-Fc conjugated to DSPE-PEG-NHS) with 1.0 mM H₂O₂ treatment exhibited excellent bioactivity consistent with that of free PD-L1-Fc, suggesting that PD-L1-Fc conjugated to DSPE-PEG-NHS did not alter the bioactivity of PD-L1-Fc, and the active site of PD-L1-Fc was well preserved.

5. It is not clear whether the ROS produced by inflammatory cell is existed inside or outside of the cell. If the ROS cleavage is happened intracellularly (as illustrated in Figure 1) after the nanoparticles are internalized, how could the PD-L1-Fc cargo be excreted outside of cell? Any data to support? In addition, nanoparticles in vivo mostly are cleared non-specifically by mononuclear phagocytic system, e.g Kupffer cells in liver and monocytes in spleen, the PD-L1-Fc bind to the NP will be cleared as well, which will not be functional. Fluorescent labeling study could not tell if the protein drugs are free in intercellular space or being trapped inside of endosomes.

Response: Reactive oxygen species (ROS) exist both intracellularly and extracellularly^[1,2], and PD-L1-Fc/Oxi- α CD NPs can be hydrolyzed by high levels of extracellular ROS in inflamed tissue, which leads to the release of PD-L1-Fc. The released PD-L1-Fc can bind to its receptors on the surface of inflammatory cells, such as PD-1, and relieve enteritis by regulating inflammatory cells and inflammatory cytokines. It is inevitable that PD-L1-Fc/Oxi- α CD NPs will be internalized by phagocytes, which may influence the function of PD-L1-Fc. However, our results demonstrated that most particles can play a role before being internalized by phagocytes. We also added the

revised content to the graphical abstract.

In addition, nanoparticles modified with polyethylene glycol (PEG) can shield the surface charge, inhibit the adsorption of serum proteins, and reduce the phagocytic ability of the mononuclear phagocytic system toward nanoparticles, thus prolonging the retention time of the nanoparticles in the circulatory system^[3].

In the previous manuscript, we used Cy5-labeled Oxi- α CD materials to fabricate Cy5-labeled PD-L1-Fc/Oxi- α CD NPs to detect the distribution of PD-L1-Fc/Oxi- α CD NPs in the major organs and various inflammatory cells. This method is not rigorous, as the Oxi- α CD NPs are prone to being hydrolyzed by ROS, which would make it impossible to detect the released PD-L1-Fc after NP hydrolysis. Thus, in the revised manuscript, we used Cy5-labeled PD-L1-Fc to fabricate Cy5-labeled PD-L1-Fc/Oxi- α CD NPs to monitor their biodistribution (Fig. 2H and Supplementary Fig. 4A-4D), and further flow cytometry analysis showed that the PD-L1-Fc released from PD-L1-Fc/Oxi- α CD NPs after hydrolysis could bind to colonic lamina propria lymphocytes (Supplementary Fig. 6), indicating that the released PD-L1-Fc acts on the surface of immune cells.

References

- [1] Stoiber W, Obermayer A, Steinbacher P, Krautgartner WD. The Role of Reactive Oxygen Species (ROS) in the Formation of Extracellular Traps (ETs) in Humans. *Biomolecules*. 2015;5(2):702-723.
- [2] Sies H, Jones DP. Reactive oxygen species (ROS) as pleiotropic physiological signalling agents. *Nat Rev Mol Cell Biol*. 2020;21(7):363-383.
- [3] Suk JS, Xu Q, Kim N, Hanes J, Ensign LM. PEGylation as a strategy for improving nanoparticle-based drug and gene delivery. *Adv Drug Deliv Rev*. 2016;99(Pt A):28-51.

Some other minor comments:

- In the cellular uptake experiment, as shown in Figure S2, the number of different immune cells is too poor to make the data unreliable, especially the dendritic cells.

Response: We isolated various immune cells from the colonic lamina propria, and in the process of isolating cells, some cells were lost or died. To clearly show the distribution of nanoparticles on immune cells, we diluted the number of cells and zoomed in 60 times for observation by laser confocal microscopy. However, because Cy5 was labeled on the Oxi- α CD NPs, we cannot observe

and quantify the binding of nanoparticles carrying PD-L1-Fc or released PD-L1-Fc to different immune cells. Therefore, in the revised manuscript, flow cytometry analysis was used to quantify the binding of PD-L1-Fc to different immune cells with human PE-conjugated Fc antibody, which can specifically react with the human Fc fragment of PD-L1-Fc (Supplementary Fig. 6 and Supplementary Fig. 7).

- In the animal experiments, the number of mice should be increased to at least five.

Response: In the revised manuscript, the number of mice was increased to six in the animal experiments.

- The figures are not uniform in size, and diagrams of supplementary information are incorrectly placed in the manuscript.

Response: We have standardized the sizes of the figures following your recommendation.

- The ordinate units of Figure 2J and 3I are wrong, they should be “pg/mL”.

Response: We have corrected these mistakes following your recommendation.

- The legends of Figure 4A-F and 5A-F are chaotic.

Response: We have corrected these mistakes following your recommendation.

REVIEWER COMMENTS

Reviewer #1 (Remarks to the Author):

In the present manuscript by Tang et al, the authors prepared ROS-responsive nanoparticles and saddled a surface-coupled Fc-fused PD-L1 protein to enable docking to the inflammatory sites in DSS-induced colitis. The authors then analysed the efficacy and biological safety of this nanosystem in the DSS colitis mouse model. Initially, the idea is quite good to deliver drugs into the micromillieu of inflammation using nanotechnologies. The basic pharmacological particle analyses of biofunction and toxicity such as zeta-potential are hopeful, but the elucidation of the mechanism of ROS-responsive system work and a possible therapeutic application in patients is only vague and presumably described. It is unclear why the so-called particles are already in the colon after 2h, although the stomach/small intestine passage takes 6h in mice. It is also unclear that the distribution of the particles is mostly and most strongly seen in the caecum.

The authors use the term nanomaterial or nanoparticle inflationarily, although the definition sets clear limits. It is the size of nanoparticles that plays a key role in their long circulation, biodistribution and clearance in the body. Typically, nanoparticles have a size between 1 and 100 nm, which allows for a longer half-life in circulation in vivo and just experiences reduced liver filtration. Furthermore, it is the small size that allows for higher intracellular uptake compared to microparticles ($1 > 1000 \mu\text{m}$), as reported by Desai et al. (1997). In the uptake study by Desai et al, Caco-2 cell line was incubated with 100-nm NPs and 1- and 10- μm microparticles and NPs had 2.5- and 6-fold higher uptake than the microparticles. The authors in the article here have a size of 118 - 221 nm, which is much larger than NPs. It is known that NPs smaller than 10 nm can easily pass through blood vessels and are excreted by the kidneys, while larger NPs can be captured by cells of the mononuclear phagocyte system (MPS), but this was not further analysed in detail.

The shape of the NPs (sphere, cube, rod, plate can be present) also plays a fundamental role in the internalisation and drug release processes. Several studies have shown that spherical particles in particular are good candidates for drug release, as they are more likely to be internalised and faster than rod-shaped NPs (see Chithrani et al., 2006). The shape may also influence their site of accumulation in different tissues of the organism: For example, cylindrical and regularly spherical NPs accumulate in the liver; disc-shaped NPs in the heart; and irregular spherical NPs in the spleen (Park et al., 2009; Devarajan et al., 2010). Here, unfortunately, nothing has been listed as an analysis of the particles in the manuscript, thus giving an unclear form/shape.

The manuscript is then unfortunately limited to a detailed description of the in vivo experiments in the DSS-mediated colitis model and does not provide a proper analysis of whether the so-called particles are really at the site of inflammation or are ingested or uptaken by epithelila cells. More research is needed here, perhaps including fluorescence microscopic preparations, and should now be more the focus of the authors. Overall, the still descriptive character of the manuscript is in my opinion not sufficient to justify a publication for the reader of Nat.Comm.

Reviewer #2 (Remarks to the Author):

In this revised manuscript, the authors have well addressed all my previous questions with additional data or detailed explanation. Therefore, I recommend the current manuscript to be accepted for publication.

REVIEWER COMMENTS

Reviewer #1 (Remarks to the Author):

In the present manuscript by Tang et al, the authors prepared ROS-responsive nanoparticles and saddled a surface-coupled Fc-fused PD-L1 protein to enable docking to the inflammatory sites in DSS-induced colitis. The authors then analysed the efficacy and biological safety of this nanosystem in the DSS colitis mouse model. Initially, the idea is quite good to deliver drugs into the micromillieu of inflammation using nanotechnologies. The basic pharmacological particle analyses of biofunction and toxicity such as zeta-potential are hopeful, but the elucidation of the mechanism of ROS-responsive system work and a possible therapeutic application in patients is only vague and presumably described. It is unclear why the so-called particles are already in the colon after 2h, although the stomach/small intestine passage takes 6h in mice. It is also unclear that the distribution of the particles is mostly and most strongly seen in the caecum.

Response: Thank you very much for your comment. Indeed, the role of the PD-1/PD-L1 axis in autoimmune diseases has received increasing attention from researchers in recent years. However, systemic administration of PD-L1 may cause severe immune suppression, ultimately resulting in severe side effects, such as severe infections or tumors. To decrease the side effects of PD-L1 for colitis therapy, we utilized ROS-responsive nanoparticles for the targeted delivery of PD-L1 to colitis lesions based on the high concentration of ROS in these lesions. In this project, the ROS-responsive nanoparticles had a core-shell structure. ROS-responsive α -cyclodextrin (Oxi- α CD), which was synthesized by conjugating 4-phenylboronic acid pinacol ester (HPAP) to the hydroxyl groups of α -cyclodextrin, forms the core of the nanoparticles. Lecithin and DSPE-PEG₂₀₀₀-NHS were used to form the shell of the nanoparticles. In addition, PD-L1-Fc was conjugated to DSPE-PEG₂₀₀₀-NHS through a classical chemical reaction. As shown in Scheme 1, the ROS-responsive unit can be degraded by H₂O₂ treatment, which leads to the degradation of the core Oxi- α CD materials, and further degradation of the nanoparticles and the release of the PD-L1-Fc protein (*ACS Macro Lett.* 2015, 4, 1220-1224).

Scheme 1. Degradation of the Oxi- α CD materials in the presence of H₂O₂.

Our team has prepared a series of ROS-responsive nanodrug delivery systems to deliver docetaxel, moxifloxacin, dexamethasone, cefpodoxime, and luteolin for the targeted treatment of various diseases (*Adv. Healthcare Mater.* 2015, 4, 69-76; *J Nanobiotechnol.* 2019, 17:103, 1-16; *Biomaterials.* 143 (2017) 93-108; *Acta Biomaterialia.* 2022, 43, 471-486; *Drug Delivery.* 2021, 28(1):1695-1708; *Biomaterials.* 2020, 232: 119730). ROS-responsive materials can be degraded in the presence of different concentrations of H₂O₂ (0.01, 0.05, 0.25, 0.5 and 1.0 mM H₂O₂) (*Adv. Healthcare Mater.* 2015, 4, 69-76). Therefore, the mechanism by which ROS-responsive NPs are degraded was discussed in our previous work. In the present work, we provide a novel strategy to achieve immunotherapy in IBD patients via PD-L1-Fc. We aimed to develop a ROS-responsive nanoplatform for the delivery of PD-L1-Fc to inflamed sites in the colon and investigated its therapeutic efficacy, mechanism and safety. Although our study is limited to only basic research, it can provide theoretical support for clinical translation. The potential application of this nanomedicine in the treatment of IBD patients has been described in the discussion of the revised manuscript.

We sincerely apologize for not emphasizing certain details carefully enough, which may have caused confusion. For example, our graphical abstract, due to the oversight of several details, could have misled readers to believe that the administration route was oral. However, in this study, the route of administration was intraperitoneal injection, not oral administration. We have made corrections to the graphical abstract to enhance clarity and avoid confusion. The reason we administered intraperitoneal injection is that we employed functional-grade recombinant PD-L1-Fc protein in this study, and the low pH in the stomach may disrupt the structure of the PD-L1 protein, leading to loss of functionality. Consequently, intraperitoneal injection was chosen as the administration route in our study. Previous studies have also reported that

therapeutic-loaded nanoparticle treatment administered by intraperitoneal injection (*J Control Release*. 2008;130(2):129-38; *Gastroenterology*. 2018;154(4):1024-1036.e9.). Previous studies have also demonstrated that the distribution of nanotherapeutics was found in the colon after 1 hour after administration via intraperitoneal injection in a mouse model (*Int J Mol Sci*. 2018; 19(1): 205). Therefore, our prepared PD-L1-Fc nanoparticles appeared in the colon after 2 h of administration because intraperitoneal injection as opposed to oral administration was used in this study, which did not alter gastrointestinal transit time.

In addition to the colon, DSS-induced inflammation can also occur in the cecum according to some studies, as indicated by increased pathological scores and changes in the microbiota and metabolites (*Front Neurosci*. 2021;15:760606; *Carbohydr Polym*. 2023;319:121180; *Gut Microbes*. 2019;10(6):696-711; *Int J Exp Pathol*. 2015;96(3):151-162; *Clin Exp Immunol*. 1998;114(3):385-391; *Clin Exp Med*. 2015;15(1):107-120; *Sci Rep*. 2017;7(1):16500.). Thus, the distribution of the particles is seen in the colon and cecum. This can be explained by the enhanced vascular permeability caused by the inflammation in the colon and cecum, which have abundant blood vessels.

The authors use the term nanomaterial or nanoparticle inflationarily, although the definition sets clear limits. It is the size of nanoparticles that plays a key role in their long circulation, biodistribution and clearance in the body. Typically, nanoparticles have a size between 1 and 100 nm, which allows for a longer half-life in circulation in vivo and just experiences reduced liver filtration. Furthermore, it is the small size that allows for higher intracellular uptake compared to microparticles ($1 > 1000 \mu\text{m}$), as reported by Desai et al. (1997). In the uptake study by Desai et al, Caco-2 cell line was incubated with 100-nm NPs and 1- and 10- μm microparticles and NPs had 2.5- and 6-fold higher uptake than the microparticles. The authors in the article here have a size of 118 - 221 nm, which is much larger than NPs. It is known that NPs smaller than 10 nm can easily pass through blood vessels and are excreted by the kidneys, while larger NPs can be captured by cells of the mononuclear phagocyte system (MPS), but this was not further analysed in detail.

Response: This is an interesting question, one that reflects the profound expertise of the reviewer in nanodrug delivery systems. We agree with the reviewer about the definition of nanoparticles

used in drug delivery systems. The size of nanoparticles is less than 100 nm, which is beneficial for enhancing the penetration and retention (EPR) effect, which allows nanoparticles to pass through blood vessels with increased permeability (specifically in tumor or inflammation sites) (*Gut*. 2011;60(7):998-1008; *Annu Rev Pathol*. 2016;11:251-281; *Lab Invest*. 2012;92(1):9-21; *Nature Nanotechnology*, 2016, 11:533-538.). In our study, an intraperitoneal injection method was used to administer PD-L1-Fc-loaded nanoparticles. The injected PD-L1-Fc nanoparticles can enter systemic circulation mainly through the vascular network of blood and lymphatic vessels in the peritoneum since the peritoneum has an absorptive function (*Techniques in the Behavioral and Neural Sciences*. 1994, 12: 46-58; *Pharm Res*. 2019;37(1):12), and eventually reach diseased sites by passing through blood vessels. Importantly, our TEM results (Fig. 2A and 2B, Supplementary Fig. 3A and 3B) revealed that the nanoparticles had a diameter less than 100 nm, which provides an opportunity for passive targeting to colitis tissues. Dynamic light scattering (DLS) measurements provide a hydrodynamic diameter that reflects the size of the particle in solution and includes coatings or surface modifications, whereas TEM depicts the actual size of the sample in the dry state (*Sci Technol Adv Mater*. 2018;19(1):732-745). Thus, DLS usually gives results that are larger than the results from TEM. This explains why the size of our nanoparticles was greater than 100 nm according to DLS measurements but was less than 100 nm according to TEM images. In addition, we further employed a scanning electron microscope to scan the nanoparticles we prepared. The results showed that the nanoparticles were spherical in shape, with most having a diameter smaller than 100 nanometers (Supplementary Fig. 1). In the revised manuscript, the particle size measured by DLS has been uniformly written as 'hydrodynamic size'.

In our study, our nanoparticles were modified with polyethylene glycol (PEG), which can shield the surface from aggregation, opsonization, and phagocytosis; inhibit the adsorption of serum proteins; reduce the phagocytic ability of the mononuclear phagocytic system toward nanoparticles; and prolong the retention time of the nanoparticles in the circulatory system (*Bioconjug Chem*. 1995;6(2):150-165; *Adv Drug Deliv. Rev*. 1995, 16;157-182; *Adv Drug Deliv Rev*. 2016;99(Pt A):28-51). Thus, our nanoparticles were able to avoid being captured by cells of the mononuclear phagocyte system (MPS) and could be effectively delivered to diseased colons.

The shape of the NPs (sphere, cube, rod, plate can be present) also plays a fundamental role in the internalisation and drug release processes. Several studies have shown that spherical particles in particular are good candidates for drug release, as they are more likely to be internalised and faster than rod-shaped NPs (see Chithrani et al., 2006). The shape may also influence their site of accumulation in different tissues of the organism: For example, cylindrical and regularly spherical NPs accumulate in the liver; disc-shaped NPs in the heart; and irregular spherical NPs in the spleen (Park et al., 2009; Devarajan et al., 2010). Here, unfortunately, nothing has been listed as an analysis of the particles in the manuscript, thus giving an unclear form/shape.

Response: Thank you for your valuable suggestion. We apologize that we did not describe the shape of the PD-L1 nanoparticles in detail in this manuscript. As shown by the TEM results, all the nanoparticles used in this manuscript were spherical (Fig. 2A and 2B, Supplementary Fig. 3A and 3B). We further employed a scanning electron microscope to scan the nanoparticles we prepared. The results showed that the nanoparticles were spherical in shape (Supplementary Fig. 1).

Spherical NPs are the most commonly used NPs. As the reviewer mentioned, the spherical structure of nanoparticles is beneficial for facilitating drug release from nanoparticles. We have described the shape of the nanoparticles in the revised manuscript. In addition, we did not discuss the influence of nanoparticle shape on targeting efficiency since other studies have discussed this topic in detail (*Cell Biol Int.* 2015; 39: 881–90; *Nat Nanotechnol.* 2017;12(1):81-89;). We focused on the therapeutic efficacy and mechanism of action of the PD-L1-Fc nanoparticles in this manuscript.

The manuscript is then unfortunately limited to a detailed description of the in vivo experiments in the DSS-mediated colitis model and does not provide a proper analysis of whether the so-called particles are really at the site of inflammation or are ingested or uptaken by epithelial cells. More research is needed here, perhaps including fluorescence microscopic preparations, and should now be more the focus of the authors. Overall, the still descriptive character of the manuscript is in my opinion not sufficient to justify a publication for the reader of Nat.Comm.

Response: Thank you for this interesting comment. Herein, we designed these nanoparticles based

on the idea that the high ROS levels at the site of inflammation lead to hydrolysis of the nanoparticles, which facilitates the release of the PD-L1-Fc drug. The released PD-L1-Fc drug can bind to PD-1 on the surface of inflammatory cells and negatively regulate immune responses, thereby relieving inflammation (Chen LP *et al.*, *Nat. Rev. Immunol.* 2004. 4:336–47; Carreno BM *et al.*, *Annu. Rev. Immunol.* 2002. 20:29–53; Song MY *et al.*, *Gut.* 2015;64(2):260-271). Because extracellular ROS may trigger the release of protein drugs from NPs, PD-L1 nanoparticles do not need to be taken up by epithelial cells because this may result in the failure of PD-L1-Fc binding to PD-1 on the membrane of inflammatory cells.

In our study, we demonstrated that the nanoparticles (PD-L1-Fc/Oxi- α CD NPs) are sensitive to ROS *in vivo* and *in vitro* (Supplementary Fig. 2A-2D, Supplementary Fig. 4 and Supplementary Fig. 8), and can be rapidly hydrolyzed by H₂O₂ to release bioactive and functional PD-L1-Fc (Fig. 2E-2F and Supplementary Fig. 2E-2H).

To further confirm that the PD-L1-Fc/Oxi- α CD NPs indeed reached the site of inflammation, Cy5-labeled Oxi- α CD (Cy5-Oxi- α CD) was synthesized and used to prepare PD-L1-Fc/Cy5-Oxi- α CD NPs (refer to the revised manuscript). PD-L1-Fc/Cy5-Oxi- α CD NPs were intraperitoneally injected into colitis mice and healthy mice. After 8 h, the mice were sacrificed, and the colons were collected. Fluorescence imaging of colon cryosections showed that Cy5 fluorescence notably accumulated in the colonic mucous lamina propria of colitis mice but not in that of healthy mice (Supplementary Fig. 7). Because enteritis causes to an abundance of inflammatory cells accumulate in the lamina propria, the above results confirmed that the NPs were indeed at the site of inflammation.

In addition, we used FITC-conjugated anti-human IgG Fc antibodies to evaluate the delivery of PD-L1-Fc by the nanoparticles. The CLSM data revealed PD-L1-Fc/Oxi- α CD NPs treatment led abundant PD-L1-Fc in the colonic mucosa of colitis mice, while only few PD-L1-Fc in the colonic mucosa of normal mice (Fig. 2J). The colonic mucosa of colitis mice treated with PD-L1-Fc or PD-L1-Fc/PLGA NPs showed a lower fluorescence signal of PD-L1-Fc than that of colitis mice treated with PD-L1-Fc/Oxi- α CD NPs (Fig. 2J and Supplementary Fig. 9A). Moreover, the results of flow cytometry analysis showed that PD-L1-Fc/Oxi- α CD NPs treatment enhanced PD-L1-Fc binding with multiple immune cells (T cells, macrophages, dendritic cells, and neutrophils) in the colonic lamina propria of the inflamed colon compared with other treatment of PD-L1-Fc

formulations (Fig. 2K, Supplementary Fig. 9B, Supplementary Fig. 10 and Supplementary Fig. 11).

In conclusion, the above results demonstrated that the PD-L1-Fc/Oxi- α CD NPs successfully reached the site of inflammation and released PD-L1-Fc.

Reviewer #2 (Remarks to the Author):

In this revised manuscript, the authors have well addressed all my previous questions with additional data or detailed explanation. Therefore, I recommend the current manuscript to be accepted for publication.

REVIEWERS' COMMENTS

Reviewer #1 (Remarks to the Author):

None